# Physical understanding of the extreme global temperature jump in 2023
J. Mex [1,2] ✉, C. Cassou [1], A. Jézéquel[1,3], S. Bony [4] & C. Deser [5]

Global surface air temperature reached unprecedented heights in early boreal fall 2023, surpassing the previous record for year-to-year temperature increase by a significant margin. We attribute most of this temperature jump to the onset and maturing stages of the 2023 El Niño, with some contributions from the North Atlantic. Using a process-based analysis from multiple observational datasets, we show that the uniqueness of the 2023 event arose from the La Niña-like ocean-atmosphere state on which it developed. This background favoured (1) a steep year-to-year increase of Sea Surface Temperature, particularly in mean atmospheric subsidence regions, reducing low-cloud cover and giving rise to a record-breaking change in the radiative budget; (2) anomalously sustained precipitation over high sea surface temperatures in the Western Pacific, fuelling unusual diabatic heating compared to canonical El Nino events. This altogether led to an exceptionally early increase in tropical tropospheric temperature in boreal fall, ultimately influencing the jump in temperature at the global scale.

In the context of continuing greenhouse gas emissions and resultant long-term global warming, new temperature records and related extreme heat waves have become more and more frequent[1]. The year 2023, however, has set a range of new records from daily to annual timescales, both globally and regionally, that were so extreme and widespread that concerns of possible and unanticipated accelerated global warming have been put forward[2].

While anthropogenic forcing undoubtedly plays a large role in the 2023 observed temperature anomalies[3], several natural external forcing mechanisms have been proposed to explain the temperature spike, such as the phasing with the 11-year solar cycle in a rising phase[4] or the January 2022 Hunga Tonga eruption[5]. Beyond greenhouse gases, it has been proposed that the reduction of sulphur emissions resulting from the 2020 regulation on fuel qualities in marine shipping[6-8]) may have also contributed to the warming by reducing the aerosol cooling effect. The relative importance of these factors is still a matter of debate; however, together they are insufficient to explain the global temperature spike in the annual, seasonal or monthly values of global surface air temperature (GSAT) that has been observed in 2023 (refs. 9–11).

Anthropogenically-forced global warming can be temporarily amplified (or attenuated) by internal climate variability, whose extreme cases can be seen as foreshadowing impacts and risks of near-future global warming levels. El Niño Southern Oscillation (ENSO) is the main interannual mode of internal variability and modelling studies suggest that it has played a major role in setting the 2023 global annual temperature record (refs. 12–15). On a monthly timescale, analyses of large simulation ensembles suggest that the observed jump in the September GSAT record can be regarded as a remarkable event with very low probability of occurrence, even at the current global warming level (refs. 9,16,17). The rarity of these types of events as simulated in models could either be an extreme expression of internal variability phenomena or be due to intrinsic model biases, including for instance, the underestimation of intraseasonal internal variability in climate models or deficiencies in capturing the diversity of ENSO dynamics (temporality, spatiality, etc.) and related global teleconnection[18]. The extreme nature of 2023 also raises the question of possible changes in the mechanisms and dynamics of internal variability in a non-stationary climate.

After a multi-year La Niña event in 2020–2022, the year 2023 saw the build-up to a moderate-to-strong El Niño event. An increase in global temperature was thus expected, but not as early as the August-to-October (ASO) early fall season, considering the canonical lagged relationship between ENSO and GSAT[10]. Using methods of non-stationary normals, Cattiaux et al.[19] confirm that the timing is of singular rarity in the observational record while also providing evidence that the jump of annual temperature in 2023 is comparable to other El Niño episodes (e.g., 1997–1998) when accounting for anthropogenically-forced global warming trends[20].

Statistical analysis of multi-model ensembles (refs. 12,13), as well as pace-maker experiments, highlight the importance of the preceding La Niña on tropical changes, but fall short in explaining the underlying mechanisms as well as the extratropical warming that characterized fall 2023. At the same

[1]LMD/IPSL, École Normale Supérieure, PSL Université Paris, Paris, France. [2]Leipzig Institute for Meteorology, Leipzig University, Leipzig, Germany. [3]École Nationale des Ponts et Chaussées, Champs sur Marne, France. [4]LMD/IPSL, Sorbonne University, CNRS, Paris, France. [5]National Center for Atmospheric Research, Boulder, CO, USA. ✉e-mail: julius.mex@uni-leipzig.de

time, several observational studies have highlighted the extremes in Earth's radiative budget (refs. 21,22) as well as the unusual tropical atmospheric pattern of the 2023 El-Niño (refs. 23,24).

Our study uses observations to connect the specific characteristics of the 2023 El Niño with the observed extreme jump in the early fall (ASO) GSAT and subsequent annual record, based on a process-based approach using observations only. We will focus on that specific early fall season, which is considered atypical for an ENSO year following Cattiaux et al.[19]. In "An extreme temperature jump in the early fall of 2023", we first analyze the contributions of the various ocean basins to the global temperature jump and quantify the importance of the Indo-Pacific Ocean. "Extreme jump in radiative budget" and "Extreme jump in tropospheric heating" then unravel two physical mechanisms that explain both the magnitude of the jump and the extreme level of the observed warming in the early fall of 2023. The implications of these findings are discussed in "Discussion and Conclusion".

## Results

### An extreme temperature jump in the early fall of 2023

The change of global marine surface air temperature (GMSAT) between two consecutive years in ASO seasons is record high between 2022 and 2023 (Fig. 1a). The onset of El Niño generally leads to a strong year-to-year global temperature jumps as observed in 1987, 1997 and 2015, but the margin by which the early fall temperature change record was broken in 2023 is remarkable. Note that cooling occurred in 1982 and 1991 despite the development of El Niño because of the dominating influence of the volcanic eruptions of El Chichon and Pinatubo, respectively. The contributions of the various ocean basins to the ASO GMSAT anomaly are assessed by considering their respective area-weighted anomalies (Fig. 1b). The Southern and North Atlantic Oceans, which have been warming in recent years due to the combination of pronounced decadal variability and human influence (refs. 25,26) have both contributed significantly to the 2023 ASO temperature anomaly. The GMSAT jump, however, results overwhelmingly from the Indo-Pacific Ocean, contributing by about 66% to the GMSAT change, which averages to 0.36 °C (29% and 38% from tropical and extratropical Indo-Pacific, respectively). Furthermore, the changes in global surface air temperature over land (GLSAT) were record high too in 2023 (Fig. S1a) and contributed about as much as GMSAT to the total jump in ASO GSAT global temperature. The land anomalies stem largely from the Tropical and Northern Midlatitudes regions (27% and 42%, respectively, see Fig. S1). Tropical oceans, and the tropical Indo-Pacific in particular, are the major sources of diabatic heating of the atmosphere (refs. 27,28), and their variability is thus of great influence for GSAT variations through planetary-scale tropical-extratropical teleconnection (refs. 29,30). We thus focus on the tropical Indo-Pacific region in the following sections.

Despite the Indo-Pacific Ocean having the strongest variance at the interannual timescale, no comparable temperature jump is observed for any of the other strong-to-moderate El Niño events since 1979 (Fig. S2). Regarding the temporal progression of the SST anomalies, the onset of the 2023 El-Niño event follows a seasonal evolution that is comparable to those of other strong El-Niño events according to the Oceanic Niño Index (ONI) (Fig. 1c), with a progressive build-up in boreal spring and maximum intensity in early winter. There is a diversity in El Niño history, with events starting in all possible phases of ENSO in the year before. The specificity of the 2023 event lies in the strong La Niña state present in 2022, with the 2009 event being the closest analogue based on ONI.

### Extreme jump in radiative budget

To understand how the preconditioning of the Indo-Pacific in 2022 impacted the evolution of SST, cloud cover, and radiative anomalies, and contributed to the observed sudden increase of tropical Indo-Pacific surface air temperatures in ASO 2023, we consider the change in SST (ΔSST) between those two years based on April-to-September (AMJJAS) means. This extended period prior to ASO, our season of interest, is chosen to capture the radiative budget anomaly contribution in the build-up of SST anomalies in ENSO events following Ceppi and Fueglistaler[31].

The change in AMJJAS SST between 2022 and 2023 is characterized by a warming in the Eastern and Central Pacific (Fig. 2a) that was both stronger in magnitude and more widespread compared to canonical El Niños (Fig. 2b, c). This specificity is attributable to the rapid switch to El Niño from a triple-dip La Niña and related strong cold anomalies along the South American coast and more broadly over the Southeastern tropical Pacific basin that were present in 2022[14]. In the central basin, the Pacific system flipped from an intense seasonal cold tongue penetrating westward at the equator around the dateline in 2022, which is typical for La Niña events[32], to a retracted one in 2023. When ΔSST is assessed as a function of climatological mean ascending and descending motion in the mid troposphere (at 500hPa,) over the Indo-Pacific domain, results show a much larger increase of SST in atmospheric subsidence regimes in 2022–2023 than during other El Niños, particularly in the regions of strongest mean subsidence (i.e. the southeastern tropical Pacific) (Fig. 2g).

Changes in SST have an impact on the Top of Atmosphere (TOA) energy budget through their influence on the lower tropospheric stability (defined as $LTS = \Theta_{700} - \Theta_{1000}$, where $\Theta$ is the potential temperature[33]). Convective adjustment makes the tropospheric temperature ($\Theta_{Tropo}$) follow a moist adiabatic profile, which is anchored to the warmest SSTs in deep convective regions, while dynamic adjustment renders the temperature horizontally uniform on a timescale of a few days to weeks (refs. 34–36). The temperature in the boundary layer ($\Theta_{BL}$), on the other hand, is more influenced by local SSTs (ref. 37). Therefore, warming of SSTs (particularly in regions of subsidence) increases the local $\Theta_{BL}$ and decreases LTS and consequently the inversion strength, favouring a decrease of low cloud cover[33] (Fig. S3a). This reduces the planetary albedo and leads to a positive radiative budget anomaly at the TOA. ENSO-driven interannual variability of lower tropospheric stability and low cloud cover, as well as the resulting modulation in shortwave cloud radiative effect, has been reported in observational and modelling studies (refs. 38–41). Recently, Fueglistaler[42] used observational records to apply the so-called pattern effect (refs. 28,43–45) to the build-up of an El Niño event to explain why the warming in the Niño3.4 region is preceded by a several-month-long positive radiative budget anomaly in subsidence regions; Ceppi and Fueglistaler[31] subsequently confirmed these results through a study combining models and observations.

Through this mechanism, the steep SST change in subsidence regimes from 2022 to 2023 (Fig. 2g) strongly impacts the change in TOA radiation budget (ΔN) between the two years (Fig. 2d), which spatially overlaps with the jump in SST (Fig. 2a). The robustness of ERA5 anomalies is verified by comparison to CERES (Fig. S4). Maximum radiative excess is located (i) in the central Pacific, (ii) along the southern flank of the climatological ITCZ at 5 °N of latitude with reinforced anomalies in the easternmost part of the basin closer to the Equator and (iii) over the mean subsiding areas to a lesser extent in absolute value. The 2022–2023 jump in ΔN is much more pronounced in the latter two domains compared with canonical El Niños (Fig. 2e, f). This is consistent with the ΔSST pattern and related increase of $\Theta_{BL}$ leading to weaker local tropospheric stability and a stronger reduction in 2022-2023 of the low-level cloud cover (Fig. S3a–c). This is especially pronounced over the regions of strongest subsidence (Fig. 2h), matching the SST behaviour (Fig. 2g). Low-level clouds also control the radiative budget over the marginal zones of the subsidence areas that are subject to variations in the sign of ω, like in the central basin and south of the ITCZ. The strongest jump in SST (Fig. 2c) and low cloud cover (Fig. S3c) along the coast of America is associated with the onset of the very extreme coastal El Niño in early 2023 (Peng et al. 2024). At the western extremity of the equatorial cold tongue, low-level clouds decreased more strongly between 2022 and 2023 than for canonical El Niño shifts, consistent with stronger ΔSST to the west of the dateline.

While an El Niño develops, convection shifts eastward from the warm pool, causing an increased high-cloud cover in the central Pacific and locally affecting ΔN. The specificity of the 2022–2023 event lies in the persistence and reinforcement of convection and related high-level cloud over the western Pacific basin (west of 170° West), which usually

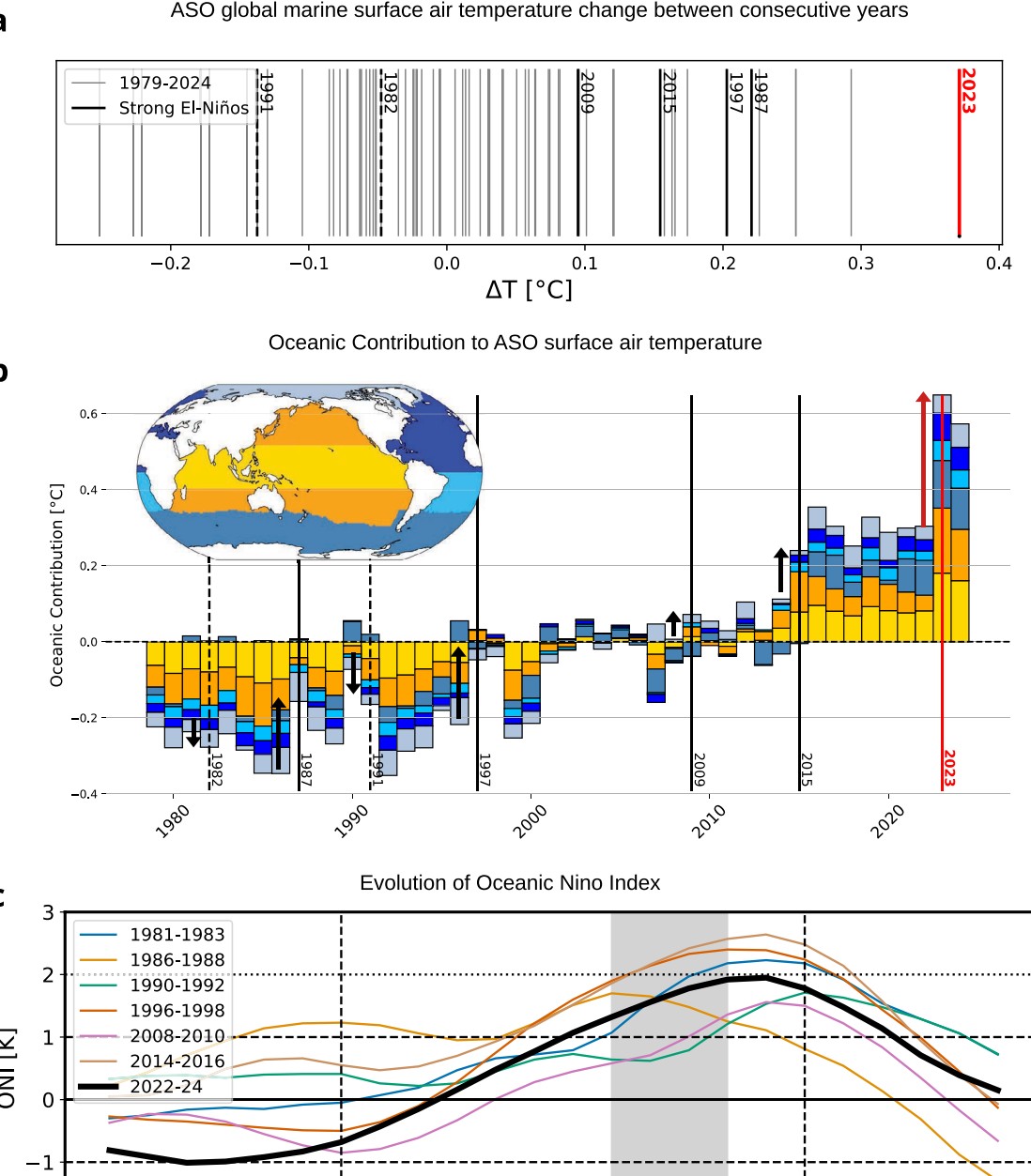

**Fig. 1 | Observed record-breaking jump of Global Marine Surface Air Temperature. a** Change of Global marine surface temperature during August-September (ASO GMSAT) between two consecutive years for the years 1979-2024 (grey), with strong El-Niño years highlighted in black (dashed for years of volcanic eruptions) and 2023 in red, all in °C (ERA5). **b** Area-weighted oceanic contributions to GMSAT-anomalies for the ASO season relative to 1991-2020 in °C (ERA5). The built-up years of strong El Niño years are highlighted in black (dashed for years of volcanic eruptions), with 2023 in red. The oceanic basins represented in different colours (upper-left map) are defined from Fay and McKinley[67]. Arrows show the size of the jump. **c** The ENSO cycle (from June-August (JJA) Year [−1] to JJA Year [+1]) assessed from Oceanic Niño Index for the El Niño selected years (light coloured solid lines) and the 2022–24 event (thick black), The ASO season Year [0] is shaded in grey.

diminishes in canonical El Niños (Fig. S3d–f). When stratified in $\omega_{500}$ deciles over the entire Indo-Pacific basins, 2022–2023 changes in high-level clouds differ from other El Niños in the deepest convective regions (first decile, where ascent is the largest) but not in subsidence zones which undergo canonical changes (Fig. 2i). Consistently, the upward branch of the Walker Circulation as delineated by the $\omega_{500} = 0$ contour (Fig. 2c, f) exhibits a smaller eastwards shift towards the central Pacific in 2022-2023 than is typically seen for previous El Niño events. The latitudinal dipole in

ΔN located off Australia-New Guinea (Fig. 2d, f) features an equatorward shift of the South Pacific Convergence Zone, evidenced by reduced high-level clouds (Fig. S3d–f) replaced by increased low-level ones (Fig. S3a–c), whose combined radiative effects lead to a strong but confined loss in radiative budget centered at 15°S-160°W.

As a result of all these features, the jump in radiative budget between 2022 and 2023, when averaged over the entire tropical Indo-Pacific domain, turns out to be the second largest since 1979, behind 2011-2012 (Fig. 2j). The

### Change in SST between April-September [-1] and April-September [0]

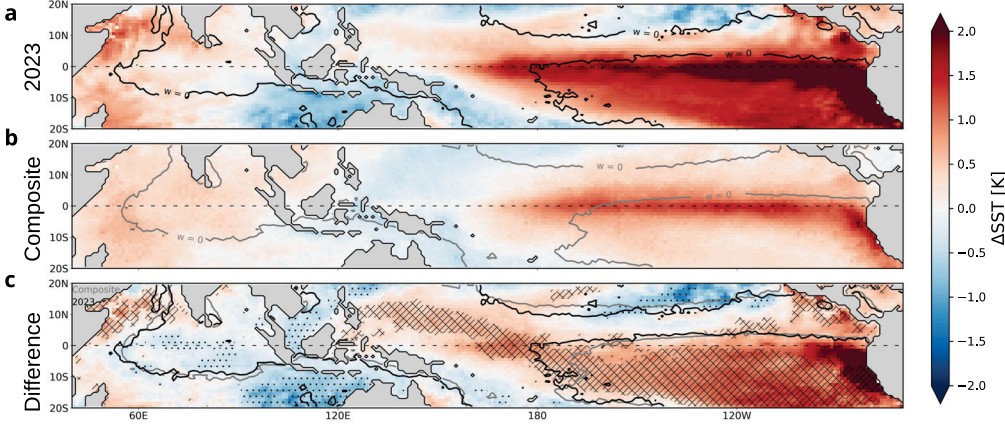

### Change in TOA Radiative Budget between April-September [-1] and April-September [0]

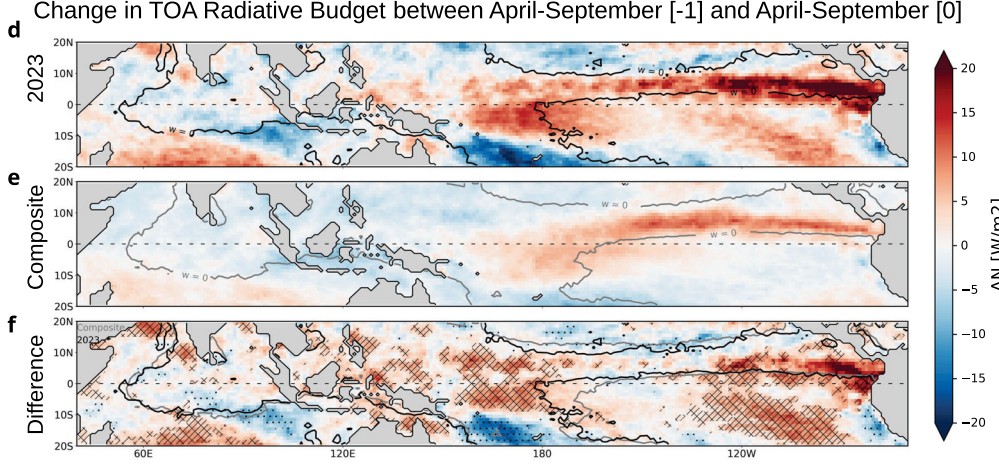

### Changes in SST, low cloud and high cloud cover grouped by vertical air motion

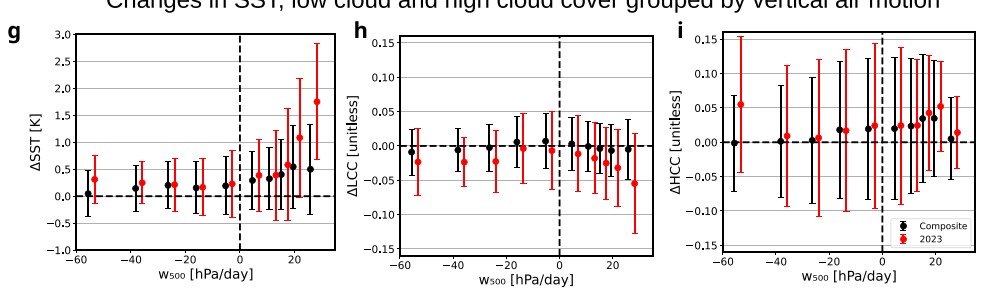

### Tropical Indo-Pacific Average of Change in TOA Radiative Budget between April-September [-1] and April-September [0]

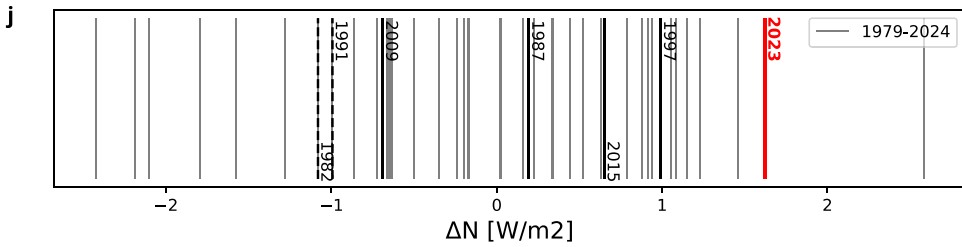

**Fig. 2 | Changes in SST and Net radiation for the 2023 El Niño built-up and the composite El-Niño.** Changes in SST leading up to ASO season, calculated as the April-to-September (AMJJAS) average anomalies of ERA5 SST for **a** 2023-2022, (**b**) Year [0] – Year [−1] for the composite year (1982, 1987, 1991, 1997, 2009 and 2015) and (**c**) the difference between 2023 and the composite event. **d**–**f** Same but for net radiation. In (**a**) and (**f**), cross-hatching is used where the 2023 anomaly is higher than all years of the composite event, and stippling where it is lower than all. The $\omega_{500} = 0$ contour is shown for 2023 (**a**, **d**), the composite (**b**, **e**) and both **c** and **f**. The collocation of changes in tropical (S20-N20) Indo-Pacific (**g**) SST, (**h**) low cloud cover (LCC), and **i** high cloud cover (HCC), defined as the AMJJAS change between Year [−1] and Year [0], binned into deciles of the climatological $\omega_{500}$ AMJJAS distribution. The error bars denote the spread as ±1σ for the composite (black) and 2023 (red). **j** Change of tropical (S20-N20), Indo-Pacific average of TOA radiative Budget between AMJJAS [−1] and AMJJAS [0] for the years 1979-2024 (grey), strong El-Niños (black) and 2023 (red) in $Wm^{-2}$ from ERA5.

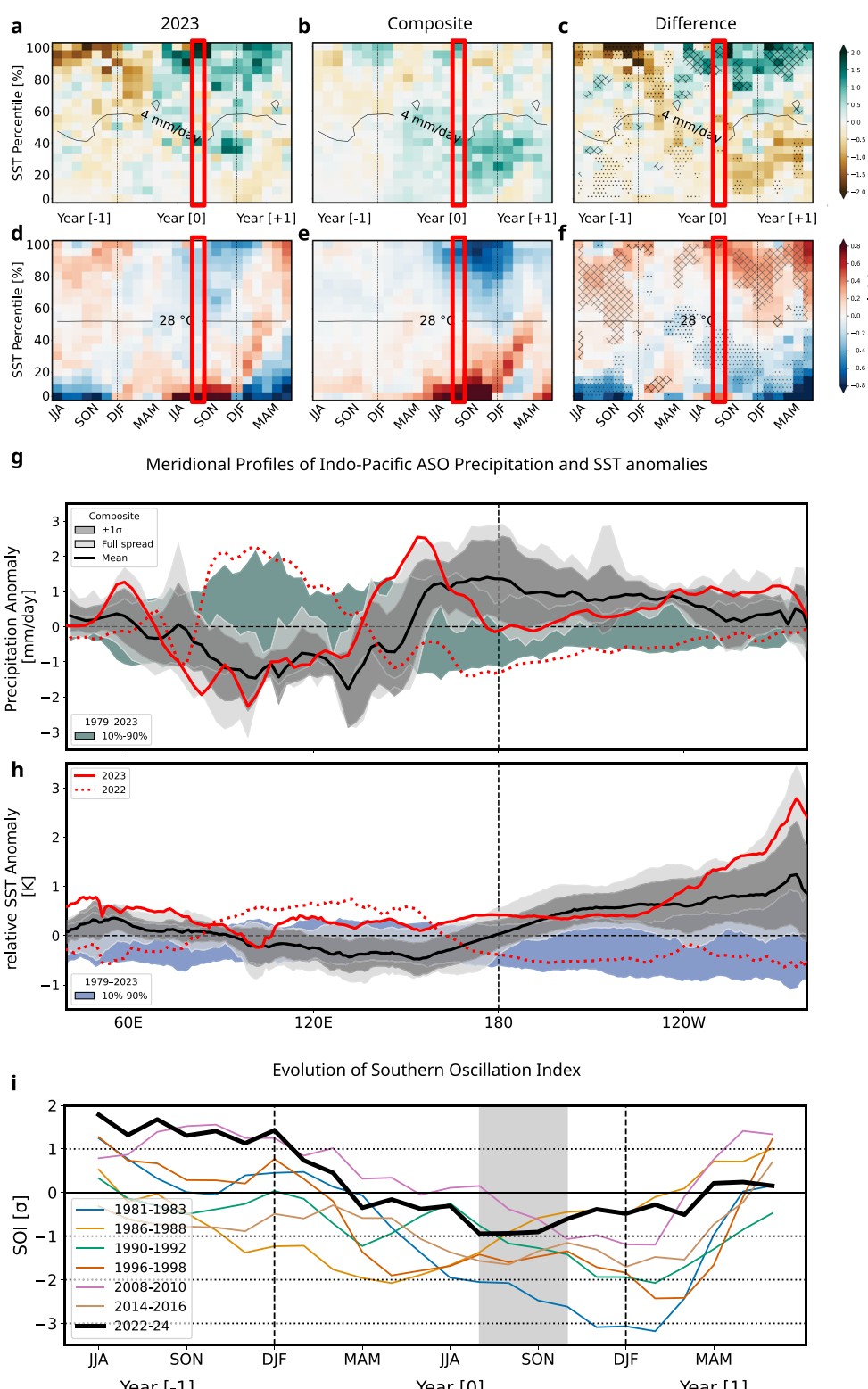

Spatial-Temporal Evolution of Indo-Pacific Precipitation and SST anomalies

positive $\Delta N$ is accompanied by a drop of about 2.5 standard deviations in low-level clouds. Such a link is consistent with the linear relationship between the two quantities, but the amplitude of the loss is the largest (Fig. S3g); the closest ENSO analogue to 2022–2023 is 1996–1997 according to this metric. By contrast, the huge drop in low-cloud cover is not associated with a large change in low tropospheric stability when averaged over the entire Indo-Pacific basin (Fig. S3h) as opposed to 1996–1997. This is due to the fact that $\Theta_{\text{Tropo}}$ and $\Theta_{\text{BL}}$ warm simultaneously in the 2022–2023 event. While many regions throughout the globe have experienced positive radiative anomalies[22], the main contribution of TOA net incoming radiation when averaged globally between consecutive AMJJAS seasons, lies in the tropical Indo-Pacific, with an area-weighted contribution of 47%. The

**Fig. 3 | Evolution of Precipitation and SST anomalies grouped by the climatological SST distribution.** Tropical (15–25 N), Indo-Pacific 3-month running mean precipitation anomalies (in mmday$^{-1}$) binned in equal area percentiles of the SST 1991-2020 climatological distribution, from coldest to warmest as in Fueglistaler 2019, using 20 bins for **a** July 2022 to June 2024. **b** composites July [Year −1] to June [Year 1] (1982, 1987, 1991, 1997, 2009, 2015) and **c** the difference between 2023 and the composite. **d–f** As **a–c** but for relative SST (mean tropical SST removed) anomalies (in Kelvin). Hatching (stippling) highlights regions where the 2023 anomalies are larger (smaller) than all of the composites. The ASO season is highlighted by the red boxes. **g** Meridional profile of the tropical (S15-N25), land-masked ASO anomalies of precipitation (in mmday$^{-1}$). Green shading shows the variability (10th to 90th percentile) for the years 1979–2024. The solid black line corresponds to the composite, with the grey (lightgrey) shading for the ±1σ (full) spread. The solid red line indicates ASO 2023, the dotted red line indicates 2022. **h** As in (**g**) for SST anomalies (in K). **i** ENSO cycle (from JJA Year [−1] to JJA Year [+1]) assessed from the Southern Oscillation Index for the El Niño composite years (light coloured solid lines) and the 2022–24 event (thick black), The ASO season Year [0] is highlighted in grey.

resulting upper ocean warming in the tropics sets the scene for precipitation and atmospheric circulation changes and related processes, which are examined subsequently.

## Extreme jump in tropospheric heating

As outlined in the introductory part of "Extreme jump in radiative budget" it is the SST in regions of deep convection that controls a large fraction of variance in tropospheric temperature through injection of anomalous diabatic heating into the tropical atmospheric column; from there, the released latent heat influences GSAT through teleconnections. To understand the temporal evolution of tropospheric temperature, we assess the spatio-temporal evolution of precipitation anomalies from 2022 to 2024 as a function of climatological SSTs ranked in percentiles (Fig. 3a–c). During an El Niño event, a reduction of precipitation usually occurs over the Indo-Pacific warm pool, where the climatological background state for atmospheric deep convection is located[46]. By contrast, rainfall increases over the lower SST deciles from early fall onwards (Fig. 3b); this see-saw reaches its maximum amplitude in boreal winter and following early spring in parallel with the SST anomalies (Fig. 3e). However, the 2022–2023 event significantly differs from canonical ENSO (Fig. 3c, f). Enhanced precipitation over middle-to-low climatological SST occurred but was considerably less pronounced and widespread in 2022–2023 compared to canonical El Niños. Over regions of climatologically high SSTs, rainfall increased from summer 2023 onwards instead of decreasing as in canonical El Niños, as also evidenced by the high cloud cover anomalies documented earlier (Fig. S3d). Illustratively, the meridional profile of the ASO tropical rainfall anomalies shows precipitation excess along the equator which remained confined to Western Pacific in 2023 whereas it is usually shifted to the Central basin for canonical El Niños (Fig. 3g). This pattern persisted until 2024, which is a clear specificity of the 2023–2024 event with respect to its strong historical counterparts (Fig. 3c).

Concurrently, SSTs dropped significantly less over the climatologically highest SSTs (Fig. 3d, f) and warmed less over the coldest ones, except at the early stage of the event associated with the development of a strong coastal El Niño[23]. In ASO 2023, SST anomalies remained warmer than average over the warm pool (Fig. 3h and S5d). Maximum positive SST anomalies were found at the easternmost part of the basin but the seasonal background ocean cooling that started in May precluded deep convection (Fig. 3g), consistent with Peng et al.[23]. At the same time, the Eastern Pacific SST warming considerably reduced low-level cloud cover and perturbed the radiative budget as documented in the previous section. Note that the gradient of tropical SST anomalies in ASO is located around 110 °W in 2023, a position that is well eastward displaced compared to about 180° for canonical El Niños. Such a longitudinal position is too far east to efficiently reduce the Walker cell, which is consistent with the overall weak response in SOI mentioned earlier (Fig. 3i). The specificity of the 2023 El Niño event to have followed La Niña years (Fig. 3h, dotted line) is well captured in Fig. 3f with positive (negative) SST anomalies over the warmer (colder) climatological SSTs. Interestingly, this pattern remained nearly unchanged in geographical structure and amplitude across the change in ENSO phases between 2022 and 2023, as also measured in SOI (Fig. 3i) with a persistently more intense Walker circulation and more convective rainfall over the Western Pacific across the entire ENSO cycle (Fig. 3g).

As a result of these atypical precipitation-to-SST configurations, the tropical atmospheric warming in 2023 started earlier in boreal summer with respect to other El Niños as assessed from potential temperature at 500 hPa, $\bar{\Theta}_{500}$ (Fig. 4a).

During a canonical El Niño, atmospheric tropical heating through anomalous convection occurs late in the course of the year, from late boreal fall to the following early spring, being phase-locked with the seasonal cycle of the SST (Fig. 3b), especially at the equator along the cold tongue. Specifically, during the boreal summer and fall seasons, positive SST anomalies occur preferentially in areas of low mean SST (non-convective), and are not associated with a significant heat source for the troposphere until late fall when SSTs are warm enough to trigger convection[42]. This explains the lag between the tropospheric temperature warming and the ONI SST anomalies. In 2023, reinforced deep convection over the warmest climatological SSTs that were even warmer than during other El Niños (Figs. 3 and S5) is hypothesized here to have produced an efficient warming of the tropical troposphere as early as late summer; this feature persisted until the end of the event.

To explain the extremes of free tropospheric temperature that occur during an El-Niño, Sobel et al.[34] introduced a measure of SST weighted by precipitation, further developed by Flannaghan et al.[47]. The improvements in using precipitation-weighted SSTs for tropospheric temperature have been confirmed and used in coupled models (refs. 39,48,49) and observational studies[50]. The relationship between tropospheric heating and precipitation-weighted SSTs (PWS)[47] is investigated now to further illustrate the specificity of the 2023 El Niño and related jump in ocean surface air temperature and GSAT as early as ASO (Fig. 1a). PWS is defined at each grid-point as the product of raw SST and precipitation rate, normalized by the tropical mean precipitation rate at the corresponding time, so that

$$PWS = \frac{\overline{SST \cdot tp}}{\overline{tp}}$$

where overbars denote the average over the tropical Indo-Pacific basins. All data are detrended, and we focus again on the ASO season.

PWS is very well correlated with $\bar{\Theta}_{500}$, with R = 0.81 (R$^2$ = 0.67). The largest values of PWS are obtained for the strong El-Niño Years [0] used in the composite (Fig. 4b); 2023 ranked 2nd in $\bar{\Theta}_{500}$ but does not significantly depart from the other ones in terms of PWS. However, it has the third largest residual in PWS with respect to the linear relationship between the two quantities when assessed over the 1979–2024 period, larger than any of the other strong El-Niño events. In order to extract which oceanic configuration acted as a "booster" for tropospheric warming beyond the linear relationship with Indo-Pacific PWS, we calculate the correlation map of the $\bar{\Theta}_{500}$/PWS residuals on the observed SST anomalies (Fig. 4c). First, the tropical north Atlantic clearly stands out with warmer SST, providing an additional source of diabatic heating. Re-evaluating the $\bar{\Theta}_{500}$/PWS relationship accounting for all tropical basins, thus including the Atlantic, improves the relationship (R = 0.86 and R$^2$ = 0.75, Fig. S6b) between the two variables and confirms the second-order but not negligible role of the tropical Atlantic Ocean. Evidence is provided in Fig. 4c) that the exceptional state of the Atlantic in 2023 has clearly boosted the Indo-Pacific induced tropospheric warming and has contributed to the record jump in $\bar{\Theta}_{500}$ in early fall. Second, positive SST anomalies over the Indo-Pacific warm-pool where most of the deep

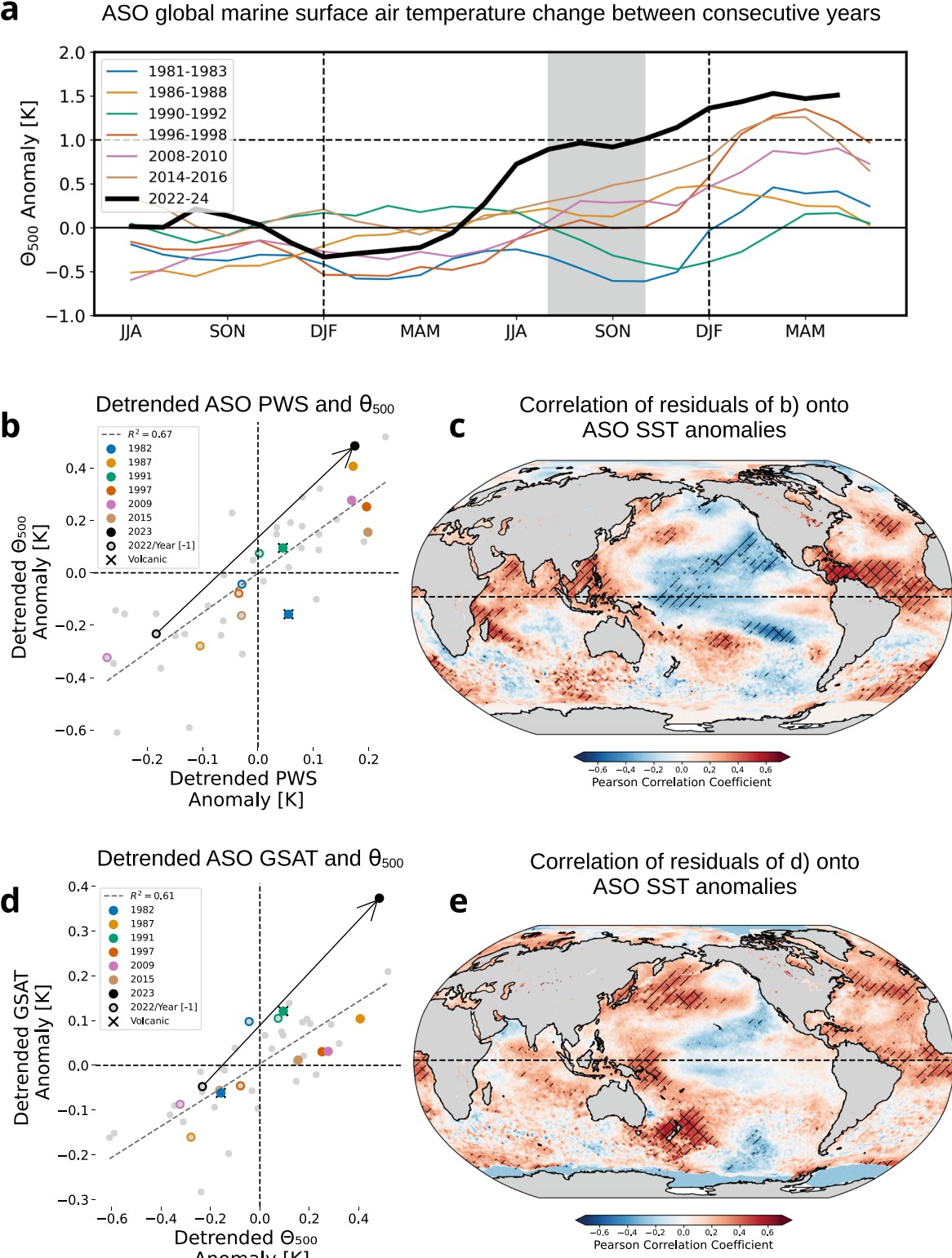

**Fig. 4 | The Evolution of tropospheric warming and SSTs in deep convective regions. a** The evolution of tropical (15S–25N) $\bar{\Theta}_{500}$ anomalies (from JJA Year [−1] to JJA Year [+1]) for the El Niño composite years (light coloured solid lines) and the 2022–24 event (thick black). The ASO season Year [0] is highlighted in grey. **b** Anomaly of tropical (15S–25N), tropospheric (500 hPa ) potential temperature for the ASO season, LOWESS-detrended, against tropical Indo-Pacific SST, LOWESS-detrended, precipitation-weighted sea surface temperature (PWS) for the years 1979–2023. The linear regression ($R^2 = 0.67$) is shown (dotted line), ENSO composite Years [0] are marked in filled colour dots, Year [−1] in empty dots. Crossed dots stand for El Niño years affected by volcanic eruptions. 2023 is marked in black, and the arrow stands for the jump between 2022 and 2023. **c** Map of Pearson correlation coefficient of the residuals of **b** and ASO SST anomalies. Diagonal hatching corresponds to areas significant at the 95th percentile, and cross-hatching for the 99th percentile based on t-statistics. **d** As in (**b**) for detrended ASO GSAT against detrended ASO $\bar{\Theta}_{500}$ anomalies ($R^2 = 0.61$). **e** As Fig. 4c for the residuals of **d** and ASO SST anomalies.

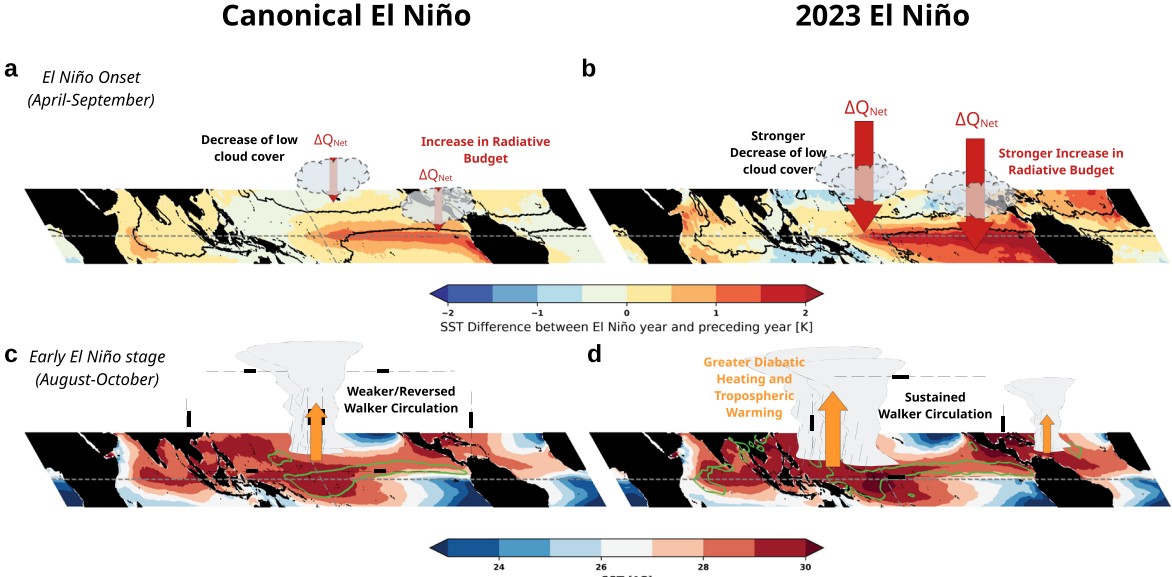

**Fig. 5 | Summary of the physical processes contributing to the year-to-year jump in Global Temperature in 2023.** Schematic overlay of the coupled ocean-atmosphere state during the Onset and early stage for canonical El Niño (**a** and **c**, respectively. and the 2023 El Niño (**b** and **d**, respectively). **a**, **b** coloured shading shows the change in SST in AMJJAS between two consecutive years when ENSO develops, with the black line being the $\omega_{500}$ equal to 0hPa s$^{-1}$ contour to differentiate between convective and subsidence areas. **c**, **d** coloured shading shows the raw SST in ASO with the green contour being the anomaly of precipitation equal to 1.5 mm day$^{-1}$ to highlight the source of anomalous diabatic heating.

convection occurs, also reinforce the tropospheric warming. The SST correlation map overall exhibits a La Niña-like pattern with large-scale teleconnection in the subtropics. This pattern largely overlaps with the regions of atypical high SSTs observed in ASO 2023, as detailed through this paper (Fig. S5f).

Finally, it is essential to mention that both PWS and $\bar{\Theta}_{500}$ were at the lower end of their distributions in 2022, notably lower than any preceding years for the other strong El Niños (Fig. 4b) over 1979–2024. This mainly explains why the jump in tropospheric temperature between two consecutive years has been the strongest in 2022–2023 (Fig. S6a).

The record jump in tropospheric temperature contributes to the jump in GSAT as the released energy within the tropics is distributed by polewards travelling Rossby waves[51]. This is confirmed by the linear relation between detrended $\bar{\Theta}_{500}$ and detrended GSAT ($R^2$ = 0.61) as shown in Fig. 4d. The 2022 to 2023 ASO changes in the two quantities deviate mildly from the regression line towards a larger jump in GSAT than to be expected from the increase in $\bar{\Theta}_{500}$ alone. The regression map of the residuals onto the ASO SST anomalies (Fig. 4e) underlines again the importance of the La Niña-like SST pattern as well as the Northern Atlantic.

## Discussion and Conclusion

This work aims, through a physical process approach, to analyze the role of ENSO, and more broadly the role of internally-driven processes, in producing the extreme jump in global surface temperature observed in 2023. We have shown that the boreal early fall season (hereafter ASO) is the most atypical aspect in relation to the 2023 El Niño lifetime when compared to other strong El Niños of the 1979-2024 period. We concentrated on this specific season and provided multiple lines of evidence that the exceptional 2023 jump in ASO temperature (Fig. 1) can be explained by the rare confluence of several physical processes in a warming climate.

The overarching factor setting the ground for their combined, strong effect lies in the preconditioning of the Indo-Pacific into a La Niña-like state. This is a specificity of the 2023-2024 El Niño with respect to the onset of other strong El Niños in recent history and results in:

1. a record year-to-year warming over the Eastern Pacific, more specifically over regions of mean subsidence during the build-up phase of ENSO events in boreal spring and summer (Fig. 2g). This is accompanied by a steep warming of boundary layer temperature there, decreasing the lower tropospheric stability and the inversion strength, therefore resulting in a sharp and widespread loss of low clouds Fig. 5b, Fig. 2h). Altogether, this led to an upsurge in the Indo-Pacific averaged radiative budget between 2022 and 2023, which was the strongest recorded for the onset months of El Niños (Fig. 2j). The emergence of an extreme coastal El Niño in spring is likely to have contributed to such a record[23]. This is coherent with the record low planetary albedo observed in 2023, as documented in Goessling et al.[22] and the atypical large Earth Energy Imbalance discussed by Minobe et al.[21] and Jiang et al.[52], to which we here provide a mechanistic explanation.

2. An atypical climate response to the El Niño build-up over the Western Pacific, both in terms of temperature, precipitation and related Walker circulation (Fig. 3). SST dropped significantly less over the warm pool (Fig. 3f) in 2023 than during other El Niños, while precipitation increased there instead of decreasing (Fig. 5d, 3c). SOI, used as a proxy for Walker cell displacement/strength, consistently remained relatively high despite strong warming over the Eastern Pacific (Fig. 3i). As shown by Peng et al.[23], the preceding, prolonged La Niña acted as a strong preconditioning factor for such specificities because of the accumulation of a large amount of warm water in the western Pacific basin by the boreal winter of 2022. Ocean heat content is record high[52,53], ensuring the stronger persistence of surface temperature anomalies over the warmest waters. In other words, the anomalous SST gradient emerging during El Niño events is located too far east in 2023 (Fig. 3h) to efficiently reduce the climatological east-west SST gradient, which canonically leads to a displacement and/or weakening of the Walker cell. Accordingly, precipitation shifted only slightly eastward between 2022 and 2023 (Fig. 3g) but remained high and was even reinforced over the warmest water of the Pacific warm pool (Figs. 3c, S5a). This also contributed to the strong radiative gain over the western Pacific basin through an atypical high cloud response (Fig. S3d).

3. An exceptional surge in tropical tropospheric warming as early as late-summer/early-fall (Figs. 5d, 4a). We show that SST variations over regions of deep convection explain about 70% of the variance of the

entire tropical atmospheric temperature through its control of the tropical moist adiabat (Fig. 4b). 2023 (i) transitioned from a cold tropical atmosphere in 2022 set by the preceding La Niña (Fig. 4b) and (ii) is atypical compared to other El Niños as it is characterized by the strongest tropospheric warming for anomalies of similar amplitude in precipitation-weighted SST, both factors leading to a record-high jump in temperature of the tropical troposphere (Fig. S6a).

All of these mechanisms, affecting the heat budget through either radiation or diabatic heating, have individually manifested during other years to a comparable extent. We argue that their atypical temporal synchronicity significantly contributed to the observed jump in early-fall global temperature. ASO is a pivotal season in the Indo-Pacific because it corresponds to the moment of the year that is influenced by the radiative anomalies developed a few months earlier over the basin and marks the beginning of the increase in diabatic heating and thus tropospheric temperature. While these processes are originally locally confined to the Indo-Pacific, their impacts, mediated through the diabatic heating and the constraints of weak temperature gradients in the tropical atmosphere, are global (Fig.4d). They contribute to explaining the extraordinary temperature changes observed in the extratropical Pacific (Fig.1b) and over land.

The large-scale background state of the Indo-Pacific has been hypothesized here to be crucial in setting the timing and extent of these mechanisms; this raises questions about the respective roles of the interannual-to-multidecadal internal modes of variability and the anthropogenically-forced trend in the preconditioning.

1. At interannual timescale, the preceding, rare triple-dip La Niña amplified the Pacific SST zonal gradient and set the stage for the jump to occur at the onset of an El Niño. Our observational analysis thus supports the importance of a preceding La Niña suggested by modelling studies (refs. 12,13). In fact, it may not be the year 2023 that was exceptional but rather the year 2022.

2. At decadal timescale, global upper-ocean heat accumulated, as a response to ever-increasing human-caused Earth Energy Imbalance, but with a large-scale regional fingerprint (refs. 21,53). Ocean heat content has risen considerably in the Western Pacific as opposed to the East; concurrently, the zonal SST gradient and the Walker cell have strengthened and low cloud cover has increased over the Eastern Pacific basin over the past decades. This again has set an atmospheric background stage for a stronger jump in radiation, with low cloud amount being initially high, close to record in 2022, and then disappearing at the onset of this El Niño in 2023, as also mentioned in Goessling et al.[22] We have shown that the large-scale SST pattern boosting the tropospheric warming, which is strongly governed by ENSO at interannual scales, projects on a La Niña-like state. It is spatially correlated at 0.42 ($p < 0.01$) to the Interdecadal Pacific mode of Variability (IPV) as defined in the 6th IPCC report[54] and reproduced in Fig. S7b). We argue here that part of the sustained precipitation in the western Pacific during the 2023 El Niño event could have been driven by the low-frequency modes of variability in the Pacific.

3. On a longer timescale, the observed Pacific SST trend is also showing a 'Niña-like' pattern (Fig. S7c). The causes of this observed trend pattern are still debated, as it is difficult to disentangle multi-decadal internally-driven variations from actual anthropogenically-driven trends over too short periods. In addition, climate models generally underestimate Pacific Decadal Variability (PDV)[55] and the anthropogenically-forced response pattern over the Pacific is still a matter of debate[56].

All together, low-frequency processes (multiyear La Niña, PDV, and trends) are poorly simulated in models (refs. 55–58). In addition, our results highlighted the importance of cloud-radiative response to SSTs and it has been shown that CMIP-class models show a large spread in the relation between low clouds and tropospheric stability while underestimating stratocumulus clouds' sensitivity to local SSTs (refs. 59,60). Therefore, both the model-observation discrepancy in the Pacific SST low-frequency variations,

forced and/or internally-driven trends, and the underestimation of stratocumulus sensitivity, could explain why climate models struggle to reproduce the observed temperature jump in 2023[16].

We have provided some evidence that the tropical North Atlantic also played a role in the early-fall jump of global temperature: (i) at the inter-annual time scale through its contribution to direct heating of the tropical troposphere, associated with an extreme phase of internal variability evaluated to be a centennial-type event[26]. Note that the Atlantic hurricane season has been ranked 4th despite El Niño, that typically results in less activity, suggesting reversed-signed teleconnection between the Pacific and Atlantic in Summer-Fall 2023[61]; (ii) through its decadal influence upon the Pacific dynamics modulating the Walker circulation and favouring a background climate projecting on negative IPV/La Niña like dynamics, characterized by sustained precipitation in the western Pacific and a reduced surface wind response over the tropical Pacific[32]. Finally, it is very likely that other factors contributed to make the ASO temperature change in 2023, and subsequently for the entire year, so extreme, as suggested from Fig.4d. While statistical models using the Green's Function Method highlight the importance of the central and western Pacific (Jiang et al., 2025), further studies are necessary to investigate and quantify the role of the tropical anomalous heat source through diabatic heating presented here, with respect to other factors that are responsible for the extreme jump in temperature observed over land (Fig. S1) and ocean (Fig. 1a) in the extratropics. For example, there are indications that cloud changes over the northern midlatitudes[22] might have had a contribution, even though they have been following a persistent, decade-long trend. Analyses of long preindustrial control simulations beyond CMIP6 historical runs, completed by dedicated model sensitivity experiments, are required to isolate and quantify the role of the different internally-driven processes documented here, as well as their interaction with the mean background state and would help to deepen our understanding of the observed 2023 jump in global temperature.

## Materials and Methods

### Data

Analyses are carried out on monthly mean ERA5[62] data at 0.25° resolution for SST, air temperature (at 2 m, 1000hPa , 700hPa and 500hPa), vertical air velocity at 500 mb, net radiation at the top of the atmosphere (TOA) and cloud cover. For precipitation, the Global Precipitation Climatology Project (GPCP[63]) data at a resolution of 2.5° is used. For consistency, we reduce our analysis to the time period over which GPCP precipitation data is available, i.e., 1979–2024. Where required, the ERA5 data is linearly interpolated onto the GPCP grid. For radiation, the Clouds and Earth's Radiation Energy Systems Energy Balanced and Filled (CERES-EBAF)[64], available from 2000 onwards, is used to evaluate the ERA5 radiation data. All anomalies are computed with respect to the 1991-2020 climatology. Detrending of data, where applicable, is done using a LOESS detrending with a bandwidth of 0.6 on the 45 years of data[65]. The Oceanic Niño Index (ONI, computed as a 3-month running SST average anomaly with respect to a 30-year climatology updated every five years, over the region 5°S-5°N, 170°W−120°W from ERA5) and the Southern Oscillation Index (SOI, computed as the standardized difference of standardized sea level pressure between Tahiti and Darwin, NOAA, https://www.ncei.noaa.gov/access/monitoring/enso/soi) are used to monitor ENSO. Except for the tropospheric temperature, which is assumed to be homogeneous under the weak gradient approximation[37], all spatial averages are computed over ocean grid points only. The term "tropical averages" refers to the 20°S-20°N domain, except when focusing on the ASO season, where we consider 15°S-25°N to account for the northward seasonal shift of the deep atmospheric convection zone.

### Composites

To make comparisons between the 2023 El Niño and past events, we constructed composite fields for strong El Niños available in the data period. Those are defined by ONI values greater than 1.5 K following the National Oceanic and Atmospheric Administration[66] guidance. This includes the 1982–83, 1987–88, 1991–92, 1997–98, 2009–10 and 2015–16 ENSO episodes.

## Data availability

ERA-5 reanalysis data are freely available in the Copernicus Climate Change Service Climate Data Store (https://cds.climate.copernicus.eu/). GPCP v2.3 precipitation data are available at https://psl.noaa.gov/data/gridded/data.gpcp.html. The CERES-EBAF dataset is available from NASA at https://asdc.larc.nasa.gov/data/CERES/EBAF/TOA_Edition4.2/. The Southern Oscillation Index is available at https://www.ncei.noaa.gov/access/monitoring/enso/soi.

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

## Acknowledgements

J.M. and A.J. have received funding from the European Union's Horizon 2020 research and innovation programme under Grant Agreement No. 101003469 (XAIDA). J.M. acknowledges funding through the Leipzig University Predoc Award. C.C. acknowledges support from IMPETUS4CHANGE, funded by the European Union's Horizon Europe research and innovation programme under grant agreement No 101081555. A.J. is supported by the ENS-PSL MACIF Chair and has received funding from state aid managed by the National Research Agency under France 2030 bearing the reference ANR-22-EXTR-0005 (TRACCS-PC4-EXTEND-ING project). S.B. acknowledges funding from the European Research Council Advanced Grant no 101098063 (MAESTRO). C.D. is funded by NSF-NCAR, a major facility sponsored by the NSF under Cooperative Agreement No. 1755088.

## Author contributions

J.M., C.C. and A.J. conceived the study. J.M. performed the analysis based on ideas of J.M., C.C., A.J., S.B. and C.D. J.M and C.C. wrote the Manuscript. J.M., C.C., A.J., S.B. and C.D. contributed to the interpretation of the results and improvement of the manuscript.

## Competing interests

The authors declare no competing Interests.
