## [Transparent Peer Review file · Communications Earth & Environment]

Physical understanding of the extreme global temperature jump in 2023

Corresponding Author: Mr Julius Mex

Version 0:

Decision Letter:

Dear Mr Mex,

Your manuscript titled "Why was the 2023 jump in global temperature so extreme?" has now been seen by 3 reviewers, whose comments are appended below. You will see that they find your work of some potential interest. However, they have raised substantial concerns that must be addressed. In light of these comments, extensive revisions will be required before we can further consider the manuscript for publication. We would, however, be interested in considering a revised version that fully addresses these serious concerns.

Specifically, we ask you to:

1. clarify the time scale and spatial domain of your analysis focus throughout your manuscript and quantify its contribution to the sudden jump of global annual mean temperature in 2023
2. discuss the contribution of mid-latitude warming and provide quantitative evidence for the link between ENSO and Indo-Pacific radiative heating.

We hope you will find the reviewers' comments useful as you decide how to proceed. If additional work allows you to either incorporate or refute these criticisms, we will be happy to look at a substantially revised manuscript. If you choose to take up this option, please either highlight all changes in the manuscript text file, or provide a list of the changes to the manuscript with your responses to the reviewers.

When resubmitting, please provide a point-by-point response to the reviewers' comments. Please submit your responses as a separate file, distinct from your cover letter where you can add responses to the Editors' comments that you do not want to be made available to the reviewers. Word files are preferred. We recommend that any figures, tables or graphs that are included in the response to reviewers are also included in the main article or Supplementary Information.

If the revision process takes significantly longer than three months, we will be happy to reconsider your paper at a later date, as long as nothing similar has been accepted for publication at Communications Earth & Environment or published elsewhere in the meantime.

Please use the following link to submit your revised manuscript, point-by-point response to the reviewers' comments with a list of your changes to the manuscript text (which should be in a separate document to any cover letter), a tracked-changes version of the manuscript (as a PDF file) and any completed checklist:

Link Redacted

Please do not hesitate to contact us if you have any questions or would like to discuss the required revisions further. Thank you for the opportunity to review your work.

Best regards,

Prof Seung-Ki Min
Editorial Board Member
Communications Earth & Environment
0000-0002-6749-010X

Alice Drinkwater, PhD
Associate Editor
Communications Earth & Environment
Consulting Editor
Communications Sustainability

EDITORIAL POLICIES AND FORMAT

If you decide to resubmit your paper, please ensure that your manuscript complies with our editorial policies and complete and upload the checklist below as a Related Manuscript file type with the revised article:

- Behavioural and social science
- Ecological, evolutionary & environmental sciences
- Life sciences

For your information, you can find some guidance regarding format requirements summarized on the following checklist: (<https://www.nature.com/documents/commsj-phys-style-formatting-checklist-article.pdf>) and formatting guide (<https://www.nature.com/documents/commsj-phys-style-formatting-guide-accept.pdf>).

REVIEWER COMMENTS:

Reviewer #1 (Remarks to the Author):

Review of Communications Earth & Environment manuscript XXX "Why was the 2023 jump in global temperature so extreme" by Mex et al.

This paper presents an observational study examining atmospheric physical processes and weather patterns in the El Niño region of the Pacific and their relationship with the extreme spike in global temperatures between 2022 and 2023.

The global temperature spike in 2023/24 has already prompted a large number of scientific papers, which are difficult to keep track of. While many of these papers have a bit of statistical focus and compare observations with climate models, this manuscript stands out as one of the most detailed investigations of El Niño development and the physical processes in that region. These processes undoubtedly played an important role in the exceptional warming spike observed in 2023.

In this regard, the authors have identified a novel angle that has not yet been fully explored in the wide body of literature addressing the 2023/24 global temperature anomaly. As far as I can tell, the analysis looks solid. Since Communications Earth & Environment has already published several papers related to this topic, I believe the journal is a suitable venue for this study. The manuscript is well structured and follows standard scientific conventions. One minor note is that in Communications Earth & Environment the Methods section is typically placed after the Discussion, though this is largely the editor's responsibility. Also, the manuscript lacked line numbers, which made providing detailed feedback more difficult.

As I am not a specialist in El Niño or Pacific climate dynamics, my comments focus mainly on the overall findings and the clarity of presentation rather than the technical details of the analysis. I hope that the other reviewers can better address the aspects that I do not cover. I have below two minor comments along with a set of line-specific suggestions, which I hope will be useful for the authors when revising the manuscript.

Minor comments:

1. The entire study focuses on oceanic areas, and specifically the Indo-Pacific region. Therefore, I was wondering about the justification of using the term “global temperature” in the title when the analysis focuses on regional, although large areas of sea surface temperatures. It should be noted that land areas also make a significant contribution to global temperature, but these are not covered in this study. Could this aspect be discussed in the Discussion section? At least I expected analysis also from land areas when I read the title.

2. The introduction is mostly clear and concise and likely contains an up-to-date overview of publications of global temperature spike 2023. However, it would help the reader if you could more explicitly highlight the research gap: what is still missing from the existing literature, and why your paper provides an important and timely contribution. Currently, you end the Introduction by stating “In this study, we investigate...”, but ideally that paragraph could precede short discussion of what previous studies have not addressed and what you consider essential to examine in this paper.

Other comments:

P3, 2nd paragraph. Typo:amay. Also, Rantanen & Laaksonen 2024 is doubled.

P3, 3rd paragraph. Modelling studies suggest vs. on model outcomes. These two word choices make the sentence a bit repetitious.

P4, 1st paragraph. I think “in climate models” is unnecessary as you talk about model biases.

P4 2nd paragraph. Cattiaux et al. 2024 without parentheses. Also, “singularity of timing” could be hard to understand? At least I do not have any idea what it means.

P6. Austral Ocean? I think this is not a well used term, or at least I needed to google what exactly is Austral Ocean.

P6 “The contribution for the temperature jump”. In this context, I’d use the term “anomaly”. In my opinion, “jump” refers to year-to-year changes while anomaly refers to the full 0.65 °C anomaly of that year. So the jump in 2023 is less than the anomaly.

Fig 1a. The explanation of dashed lines is not given in the caption.

P10, 1st paragraph. Ascendance, maybe ascent is a better word here.

P10, 2nd paragraph. Fig.S2g should be S3g I guess?

Fig. 2 caption. El Niño. It has been written with El Nino also elsewhere so please check this throughout the manuscript.

P19. In the 3rd part, you refer to Fig. 5d which is probably a typo.

Fig. S2 is the same as Fig. 1a.

References: Gyuleva et al. 2025 is now published: <https://agupubs.onlinelibrary.wiley.com/doi/10.1029/2025GL115270>

Reviewer #2 (Remarks to the Author):

The core question addressed in this study is: What drove the exceptional 2023 global temperature jump, particularly the extreme warming in boreal early fall (August-October, ASO)?

The authors emphasized the dominant role of the 2023 El Niño and Indo-Pacific Preconditioning. The 2023 temperature jump is primarily attributed to the unique physical characteristics of the 2023 El Niño’s onset and maturation, combined with North Atlantic influences. A key prerequisite is the La Niña-like ocean-atmosphere background state in the Indo-Pacific, which distinguishes the 2023 El Niño from historical strong El Niños.

Major comments:

1. Focus on 2023 early autumn temperature jump and optimize article structure & content relevance

I strongly recommend restructuring this article to focus sharply on the interannual changes in boreal early fall (ASO season) and center the discussion on the causes of the extreme interannual temperature jump during this specific period. The title may be revised as: Why was the jump in global temperature in early autumn 2023 so extreme? According to the title (“jump”), the paper should focus on the difference between 2023 ASO and 2022 ASO instead of 2023 ASO itself. Currently, the article contains excessive scattered content, making it difficult for readers to follow the core logic—especially when trying to grasp the key drivers of the 2023 temperature anomaly. For example, Fig.1b is not directly related to “jump”.

2. The significant contribution of the tropical Indo-Pacific Ocean to the jump should be confirmed.

As mentioned in the last paragraph in the Introduction, this study first analyzes the contributions of the various basins to the global temperature jump and quantify the importance of Indo-Pacific Ocean. It seems that the importance (contribution) of the Indo-Pacific serves as the foundation for subsequent research. However, the range of the Indo-Pacific Ocean in Section 3.1 is not clear, which seems to be different with the tropical Indo-Pacific in the subsequent Sections. Is the Indo-Pacific Ocean the orange area (including subtropical regions) in the figure? However, after that, the paper mainly focuses on the tropical

region. The significant contribution of the tropical Indo-Pacific Ocean should be confirmed. Figure 1b is not enough to support this point.

Minor comments:

L25-29: These sentences are not easy to understand. I hope their wording can be improved to make it easier for the readers to grasp the key points, especially the causalities within them.

L37: has -> have

L73: (Cattiaux et al.,2024)-> Cattiaux et al.(2024)

L78: El Nino-> El Niño

L81-83: This part is much clearer than L25-29 in the abstract. I suggest the abstract may be revised according to the logic of L25-29.

L89: whether "at surface" here is equal to "at 2m"?

L90:Some papers indicated that there are some issues with the TOA radiation of ERA5 (e.g., Fig. 1 of Allan and Merchant 2025), which is not consistent with the observation.

Allan R. P. and C. J Merchant (2025) Reconciling Earth's growing energy imbalance with ocean warming. Environ. Res. Lett. 20 044002 DOI 10.1088/1748-9326/adb448

L135: The contribution for ... -> The contribution to ...

L135-136: It may be revised as :The temperature jump, however, results overwhelmingly from the Indo-Pacific Ocean

Figure 1: I was confused about Fig.1a and 1b. The ASO land-masked surface air temperature change between 2023 and 2022 is about 0.58 (roughly estimated). Some explains should be added to clarify the calculation of the difference. Whether it means 2023 minus 2022? While Figure 1b shows that the total oceanic temperature is about 0.28, the difference between 2022 and 2023 should be about 0.4.

L164-165: The amplitude may be reduced in the composite result. I suggest "canonical" -> composite

Figure 2g-2i: I am curious about the calculation for the composite. I wonder whether the distributions of the red lines here are calculated by the samples (1982, 1987, 1991, 1997, 2009 and 2015) or their average. In addition, the accurate values for the x-axis bins are not clear.

L246: indo-pacific -> Indo-Pacific

Figure S3: El Nino-> El Niño. Please add more explanation about the scatters in Figure S3g-h.

L327-334 : Since the PWS is important for this study. More explanations about the physical meaning of PWS should be added.

L385: a Niña-like -> a La Niña-like

L407-409:The causal relationship implied in this sentence is unclear: Consequently, the anomalous SST gradient is located too far east in 2023 (Fig.3h) to efficiently displace or weaken the Walker circulation.

L413: The sentence is not complete: An exceptional surge in tropospheric warming as early as late-summer/early-fall.

L418: the second point: it transitioned from a cold tropical atmosphere in 2022 set by the preceding La Niña (Fig.4b). Whether this point is also atypical compared to other El Niño event?

Reviewer #3 (Remarks to the Author):

This paper analyzes the factors behind the extreme global mean surface temperature (GMST) in 2023, with a primary focus on the Indo-Pacific warming. Overall the paper is well-written. While the contribution of the tropical Pacific itself has been discussed in more sophisticated ways in previous studies, including those using climate model ensembles and tropical Pacific pacemaker experiment, this study presents a novel mechanism in which the 2023 El Niño together with La Niña-like background state drives the temperature jump through cloud radiation. This point is interesting and, in my view, merits publication. However, the neglect of the strong midlatitude land warming in 2023 and the lack of quantitative assessment of radiative heating reduce the persuasiveness of this paper as an explanation for the 2023 GMST anomaly. Therefore, I would like to request major revision.

Comment 1:

The link between ENSO and radiative fluxes is rather modest (Ceppi and Fueglistaler 2021) in contrast to the well-established strong relationship between ENSO and GMST (Kosaka and Xie 2013; Xie et al. 2025). So it is generally considered that the strong correlation between ENSO and GMST arises mainly through energy redistribution within the

Earth. Therefore stronger quantitative evidence is needed to demonstrate that Indo-Pacific radiative heating associated with ENSO significantly impacts local SST and, consequently, GMST.

Comment 2:

The 2023 GMST jump was pronounced over extratropical land regions (Xie et al. 2025). However, this study focuses exclusively on sea surface temperature, which makes the explanation for the GMST jump incomplete. Tropical Pacific SST pacemaker experiment reproduces tropical surface warming reasonably well but fail to capture extratropical warming (Xie et al. 2025). This inconsistency suggests that the GMST jump cannot be fully explained by the tropical effect.

Comment3:

The connection between the tropospheric warming discussed in Section 3.3 and GMST is not clear. In relation to Section 3.2, the tropospheric warming could increase low clouds. It would be helpful if the role of Section 3.3 could be clarified.

Comment4:

From the title and abstract, the paper seems to study the 2023 temperature. But the paper mostly focuses on the process for the Indo-Pacific changes in ASO. It is unclear if this picture contributes to the annual mean temperature.

Specific comments:

L157: "Radiative forcing" sounds like external forcing. Perhaps it should be rephrased.

L184: typo

L205: Despite the extreme coastal Niño, low cloud change along the Peruvian Coast looks small.

L231: LTS has already been defined

L233: Suggest splitting this long sentence

L308: remove ","

Fig. S6: typo in the title

Reference:

Ceppi and Fueglistaler (2021, <https://doi.org/10.1029/2021GL095261>)

Kosaka and Xie (2013, <https://doi.org/10.1038/nature12534>)

Xie et al. (2025, <https://doi.org/10.1038/s41612-025-01006-y>)

** Visit Nature Portfolio's author and referees' website at <http://www.nature.com/authors> for information about policies, services and author benefits**

Communications Earth & Environment is committed to improving transparency in authorship. As part of our efforts in this direction, we are now requesting that all authors identified as 'corresponding author' create and link their Open Researcher and Contributor Identifier (ORCID) with their account on the Manuscript Tracking System prior to acceptance. ORCID helps the scientific community achieve unambiguous attribution of all scholarly contributions. You can create and link your ORCID from the home page of the Manuscript Tracking System by clicking on 'Modify my Springer Nature account' and following the instructions in the link below. Please also inform all co-authors that they can add their ORCID to their accounts and that they must do so prior to acceptance.

Version 1:

Decision Letter:

Dear Mr Mex,

Your manuscript titled "Why was the 2023 jump in global temperature so extreme?" has now been seen by our reviewers, whose comments appear below. In light of their advice we are delighted to say that we are happy, in principle, to publish a suitably revised version in Communications Earth & Environment.

We therefore invite you to revise your paper one last time to address the remaining concerns of our reviewers. At the same time we ask that you edit your manuscript to comply with our format requirements and to maximise the accessibility and

therefore the impact of your work.

EDITORIAL REQUESTS:

****Please take care to match our formatting and policy requirements. We will check revised manuscript and return manuscripts that do not comply. Such requests will lead to delays. ****

SUBMISSION INFORMATION:

OPEN ACCESS:

Communications Earth & Environment is a fully open access journal. Articles are made freely accessible on publication. For further information about article processing charges, open access funding, and advice and support from Nature Portfolio, please visit <https://www.nature.com/commsenv/open-access>

Link Redacted

Best regards,

Alice Drinkwater, PhD
Associate Editor
Communications Earth & Environment
Consulting Editor
Communications Sustainability

REVIEWERS' COMMENTS:

Reviewer #1 (Remarks to the Author):

I thank the authors for addressing my relatively modest comments. On my behalf, I can now recommend publication. Please note that there is a brand new paper on September 2023 global jump which you may want cite in your paper: <https://www.nature.com/articles/s43247-026-03178-8>

Reviewer #2 (Remarks to the Author):

It is a pleasure to review this manuscript again. Following the authors' revisions, I now have a better understanding of the entire work, and I only have a few minor comments and suggestions.

L88: El-Niño -> El Niño

L119-120: We thus focus on the Indo-Pacific region in the following sections. -> We thus focus on the tropical Indo-Pacific region in the following sections.

L228-231: Based on my understanding, Figure 2 indicates that changes in low clouds over certain regions still follow the constraining relationship of LTS, but this relationship may not hold at the regional average level. Is my understanding correct?

L312: ofanomalous ->of anomalous

L341-342: If the relationship between delta PWS and delta theta is directly calculated, would it better illustrate the conclusions of the paper without the need for detrending? It's just a suggestion.

L392: Niña-like-> La Niña-like

L476-477: The spatial pattern of the El Niño events (double centers) seems to be a factor to the global surface temperature (Geng et al. 2024; Jiang et al. 2025).

L529-531: Recent studies have quantified the contributions of different basins to GSAT and GLSAT (Jiang et al. 2025).

Reference

Geng, X., Kug, J.S., Shin, N.Y. et al. On the spatial double peak of the 2023–2024 El Niño event. *Commun Earth Environ* 5, 691 (2024). <https://doi.org/10.1038/s43247-024-01870-1>

Jiang, N., Zhu, C., McPhaden, M.J. et al. Atypical warming pattern of strong 2023-24 El Niño boosts global temperatures to new 1.5 °C record. *Commun Earth Environ* 6, 1012 (2025). <https://doi.org/10.1038/s43247-025-02971-1>

Reviewer #3 (Remarks to the Author):

This is the second review of the manuscript. I thank the authors for carefully addressing the previous comments. I have several comments, which I believe can be addressed by modifying the text.

Section 2.3: This section starts with a detailed discussion of precipitation and tropospheric heating, whose roles in GSAT have not been clearly mentioned earlier in the manuscript. This is a point where readers may easily get lost. To guide readers, it would be helpful to clarify at the beginning of the subsection how these processes are linked to the previous subsection and how they contribute to the GSAT jump.

Section 3: The latter part of this section (L469-) is excessively long and involved, despite being speculative. I suggest shortening this part. Analysis of GLSAT should be in the Results section.

L386: It is not obvious that tropical heating drives extratropical warming. In fact, the ENSO influence on GSAT is dominated by surface warming in the tropics (Wang et al., 2025; Xie et al., 2025). The high correlation between GSAT and tropical tropospheric temperature is likely tied to surface warming in the tropics, rather than extratropical responses.

L118: As noted above, tropical influence on extratropical temperature is not the major factor for GSAT variability.

Fig. 4: There are two panels labeled (d).

L356: Typo.

L388: Alexander et al., 2002.

L442: Item 4 does not exist.

L465: I don't see any evidence for this.

L580, and some other references: Incorrect indent.

References:

Wang et al. 2017: Global Influence of Tropical Pacific Variability with Implications for Global Warming Slowdown. <https://doi.org/10.1175/JCLI-D-15-0496.1>

Xie et al. 2025: What made 2023 and 2024 the hottest years in a row? <https://doi.org/10.1038/s41612-025-01006-y>

** Visit Nature Portfolio's author and referees' website at <http://www.nature.com/authors> for information about policies, services and author benefits**

Reviewer #1 (Remarks to the Author):

Review of Communications Earth & Environment manuscript XXX “Why was the 2023 jump in global temperature so extreme” by Mex et al.

This paper presents an observational study examining atmospheric physical processes and weather patterns in the El Niño region of the Pacific and their relationship with the extreme spike in global temperatures between 2022 and 2023.

The global temperature spike in 2023/24 has already prompted a large number of scientific papers, which are difficult to keep track of. While many of these papers have a bit of statistical focus and compare observations with climate models, this manuscript stands out as one of the most detailed investigations of El Niño development and the physical processes in that region. These processes undoubtedly played an important role in the exceptional warming spike observed in 2023.

In this regard, the authors have identified a novel angle that has not yet been fully explored in the wide body of literature addressing the 2023/24 global temperature anomaly. As far as I can tell, the analysis looks solid. Since Communications Earth & Environment has already published several papers related to this topic, I believe the journal is a suitable venue for this study. The manuscript is well structured and follows standard scientific conventions. One minor note is that in Communications Earth & Environment the Methods section is typically placed after the Discussion, though this is largely the editor’s responsibility. Also, the manuscript lacked line numbers, which made providing detailed feedback more difficult.

As I am not a specialist in El Niño or Pacific climate dynamics, my comments focus mainly on the overall findings and the clarity of presentation rather than the technical details of the analysis. I hope that the other reviewers can better address the aspects that I do not cover. I have below two minor comments along with a set of line-specific suggestions, which I hope will be useful for the authors when revising the manuscript.

First of all, we would like to thank Rev#1 for his/her very constructive comments and positive evaluation of our paper. Rev#1 raised important concerns regarding the contribution of the tropical Indo-Pacific to the global temperature jump as well as the lacking discussion about land areas. Following Rev#1’s concerns, we have added related analyses and linked them with our findings, which considerably improved the manuscript. Below are our point-by-point responses to the Rev#1’s concerns.

Minor comments:

1. The entire study focuses on oceanic areas, and specifically the Indo-Pacific region. Therefore, I was wondering about the justification of using the term “global temperature” in the title when the analysis focuses on regional, although large areas of sea surface temperatures. It should be noted that land areas also make a significant contribution to global temperature, but these are not covered in this study. Could this aspect be discussed in the Discussion section? At least I expected analysis also from land areas when I read the title.

Following Rev#1, a discussion of the relative contribution of the tropical Indo-Pacific in the observed jump has been added (L169ff):

“The GMSAT jump, however, results overwhelmingly from the Indo-Pacific Ocean, contributing by about 66% to the GMSAT change, which averages to 0.36°C (29% and 38% from tropical and extratropical Indo-Pacific).”

Figure 1b) was revised to reflect the focus on the Indo-Pacific. (see reviewed Manuscript)

Even though the tropical Indo-Pacific oceanic domain is limited in surface, its influence on the entire planet is dominant because it is the main source of diabatic heating on average and its variation of the latter explains a large fraction of the global temperature variation (Graham and Barnett, 1987; Brown et al., 2015; Xie et al., 2025).

This justifies the focus on this region throughout our study. To clarify this, we have added following sentence in Section 2.1 (L171ff):

“Tropics and the tropical Indo-Pacific in particular, are the major sources of diabatic heating of the atmosphere (Graham and Bennett, 1987; Zhang et al., 2023) and their variability is thus of great influence for GSAT variations through planetary-scale tropical-extratropical teleconnection (Brown et al., 2015; Xie et al., 2025). We thus focus on the Indo-Pacific region in the following sections.”

Furthermore, we have added a paragraph in Section 2.3 to underscore the mechanism and included an analysis of the link between tropospheric temperature and GSAT, for which Figure 4 has been supplemented by panels d) and e) (see reviewed Manuscript).

“The record jump in tropospheric temperature contributes to the jump in GSAT as the released energy within the tropics is distributed by polewards travelling Rossby waves (Alexander et al, 2022). This is confirmed by the linear relation between detrended 500 and detrended GSAT ($R^2 = 0.61$) as shown in Fig.4d. The 2022 to 2023 ASO changes in the two quantities deviates mildly from the regression line towards a larger jump in GSAT than to be expected from the increase in 500 alone. The regression map of the residuals onto the ASO SST anomalies (Fig.4e) underlines again the importance of the Niña-like SST pattern as well as the Northern Atlantic. “

Nonetheless, we agree that one can not conclusively argue that the described mechanisms are the sole drivers of the observed temperature jump and that in particular land regions show a remarkable temperature change, too. To quantify these changes, we have added an analysis analogous to the one in section 2.1 for global surface air temperature over land (GLSAT). The respective figures has been added to the supplementary materials (Fig.S7a,b, see reviewed Manuscript) and the land contributions as well as other possible factors have been added to the discussion (L607ff):

“Finally, it is very likely that other factors contributed to make the ASO temperature change in 2023, and subsequently for the entire year, so extreme as suggested from Fig.4d. It is indeed important to note that changes in global surface air temperature over land (GLSAT), were record high too in 2023 (Fig. S7a) and contributed about as much as with GMSAT, to the total jump in ASO GSAT global temperature. The land anomalies stem largely from the Tropical and Northern Midlatitudes regions (27% and 42%, respectively, see Fig.S8). Further studies are necessary to investigate and quantify the role of the tropical anomalous heat source through diabatic heating presented here, with respect to other extratropical factors that are responsible for the extreme jump in temperature observed over land (Fig.S7) and ocean (Fig.1a) in the extratropics. For example, there are indications that cloud changes over the northern midlatitudes (Goessling et al., 2024) might have had a contribution, even though they have been following a persistent decade-long trend. In the same vein, the role of the Pacific Decadal Variability in explaining the strong warm anomalies along a broad midlatitude band ranging from 25N to 50N while El Niño was developing in 2023 (Fig.1b) is worth investigating.”

2. The introduction is mostly clear and concise and likely contains an up-to-date overview of publications of global temperature spike 2023. However, it would help the reader if you could more explicitly highlight the research gap: what is still missing from the existing literature, and why your paper provides an important and timely contribution. Currently, you end the Introduction by stating “In this study, we investigate...”, but ideally that paragraph could precede short discussion of what previous studies have not addressed and what you consider essential to examine in this paper.

Thank you for this remark, we hope to have highlighted the research ap we are addressing more clearly like so (L95ff):

“After a multi-year La Niña event in 2020-2022, the year 2023 saw the build-up to a moderate-to-strong El Niño event. An increase in global temperature was thus expected, but not as early as the August-to-October (ASO) early fall season considering the canonical lagged relationship between ENSO and GSAT (Schmidt 2024). Using methods of non-stationary normals, Cattiaux et al. (2024) confirm that the timing isof singular rarity in the observational record while also providing evidence that the jump of annual temperature in 2023 is comparable to other El Niño episodes (eg. 1997-1998) when accounting for anthropogenically-forced global warming trends (Forster et al. 2024).

Statistical analysis of multi-model ensembles (Raghuraman et al. 2024; Gyuleva, Knutti, and Sippel, 2025), as well as pace-maker experiments highlight the importance of the preceding La Niña on tropical changes, but fall short in explaining the underlying mechanisms as well as the extratropical warming that characterized fall 2023. At the same time, several observational studies have highlighted the extremes in Earth's radiative budget (Minobe et al., 2025; Goessling et al., 2024) as well as the unusual tropical atmospheric pattern of the 2023 El-Niño (Peng et al., 2024; Zhang et al., 2025).

Our study uses observations to connect the specific characteristics of the 2023 El-Niño with the observed extreme jump in the early fall (ASO) GSAT and subsequent annual record, based on a process-based approach using observations only. We will focus on that specific early fall season which is considered as atypical for an ENSO year following Cattiaux et al. (2024). [...]

Other comments:

P3, 2nd paragraph. Typo:amay. Also, Rantanen & Laaksonen 2024 is doubled.

Corrected

P3, 3rd paragraph. Modelling studies suggest vs. on model outcomes. These two word choices make the sentence a bit repetitious.

Corrected

P4, 1st paragraph. I think "in climate models" is unnecessary as you talk about model biases.

Removed

P4 2nd paragraph. Cattiaux et al. 2024 without parentheses. Also, "singularity of timing" could be hard to understand? At least I do not have any idea what it means.

Changed to 'Using methods of non-stationary normals, Cattiaux et al. (2024) confirm the timing to be of singular rarity in the observational records [...]

P6. Austral Ocean? I think this is not a well used term, or at least I needed to google what exactly is Austral Ocean.

Changed to 'Southern Ocean'

P6 "The contribution for the temperature jump". In this context, I'd use the term "anomaly". In my opinion, "jump" refers to year-to-year changes while anomaly refers to the full 0.65 °C anomaly of that year. So the jump in 2023 is less than the anomaly.

Changed to focus on the jump explicitly (see above).

Fig 1a. The explanation of dashed lines is not given in the caption.
Added

P10, 1st paragraph. Ascendance, maybe ascent is a better word here.

Done

P10, 2nd paragraph. Fig.S2g should be S3g I guess?
Done

Fig. 2 caption. El Niño. It has been written with El Nino also elsewhere so please check this throughout the manuscript.

Done

P19. In the 3rd part, you refer to Fig. 5d which is probably a typo.

Done

Fig. S2 is the same as Fig. 1a.

Thank you for catching this mistake. Corrected.

References: Gyuleva et al. 2025 is now published:
<https://agupubs.onlinelibrary.wiley.com/doi/10.1029/2025GL115270>

Done

References:

Stephens et al. (2015, <https://doi.org/10.1002%2F2014RG000449>)

Loeb et al. (2012, <https://doi.org/10.1007/s10712-012-9175-1>)

Graham and Barnet (1987, <https://doi.org/10.1126/science.238.4827.657>)

Brown et al. (2015, <https://doi.org/10.1002/2014JD022576>)

Xie et al. (2025, <https://doi.org/10.1038/s41612-025-01006-y>)

Alexander et al. (2002, [https://doi.org/10.1175/1520-0442\(2002\)015<2205:TABTIO>2.0.CO;2](https://doi.org/10.1175/1520-0442(2002)015<2205:TABTIO>2.0.CO;2))

We again thank the reviewer for his/her comment which helps considerably improve the original manuscript.

Reviewer #2 (Remarks to the Author):

The core question addressed in this study is: What drove the exceptional 2023 global temperature jump, particularly the extreme warming in boreal early fall (August-October, ASO)?

The authors emphasized the dominant role of the 2023 El Niño and Indo-Pacific Preconditioning. The 2023 temperature jump is primarily attributed to the unique physical characteristics of the 2023 El Niño's onset and maturation, combined with North Atlantic influences. A key prerequisite is the La Niña-like ocean-atmosphere background state in the Indo-Pacific, which distinguishes the 2023 El Niño from historical strong El Niños.

Major comments:

1. Focus on 2023 early autumn temperature jump and optimize article structure & content relevance

I strongly recommend restructuring this article to focus sharply on the interannual changes in boreal early fall (ASO season) and center the discussion on the causes of the extreme interannual temperature jump during this specific period. The title may be revised as: Why was the jump in global temperature in early autumn 2023 so extreme? According to the title ("jump"), the paper should focus on the difference between 2023 ASO and 2022 ASO instead of 2023 ASO itself. Currently, the article contains excessive scattered content, making it difficult for readers to follow the core logic—especially when trying to grasp the key drivers of the 2023 temperature anomaly. For example, Fig.1b is not directly related to "jump".

We thank the reviewer for their comments. Reviewer #2 has emphasised the need to be clearer in the studies focus with regard to the time period investigated and to show the contribution of the focus area (i.e. the tropical Indo-Pacific to the global temperature change). The remarks have helped to, in our eyes, improve the manuscript significantly.

We agree that the focus on the season (ASO) and the change (in contrast to the anomaly) should be apparent from the abstract. We thus underline the seasonal focus in the abstract:

"The record of year-to-year temperature increases was surpassed by a significant margin, especially in early boreal fall. We attribute the majority of this seasonal jump to the onset and maturing stages of the 2023 El Niño event."

Arrows have been added in Figure 1b) to visually underline the temperature change as the target variable of our study (see revised Manuscript).

We have further streamlined the reported result in the abstract (L28ff) and adapted the introduction to clearer outline the research gap addressed in our study (L104ff). We hope that this, alongside the changes outlined in the response to the following comment, help clarify our papers core logic.

We have however decided to keep the original title, as we think that with the received abstract, the focus on the season and jump is apparent and has furthermore been underlined further in the introduction.

2. The significant contribution of the tropical Indo-Pacific Ocean to the jump should be confirmed.

As mentioned in the last paragraph in the Introduction, this study first analyzes the contributions of the various basins to the global temperature jump and quantify the importance of Indo-Pacific Ocean. It seems that the importance (contribution) of the Indo-Pacific serves as the foundation for subsequent research. However, the range of the Indo-Pacific Ocean in Section 3.1 is not clear, which seems to be different with the tropical Indo-Pacific in the subsequent Sections. Is the Indo-Pacific Ocean the orange area (including subtropical regions) in the figure? However, after that, the paper mainly focuses on the tropical region. The significant contribution of the tropical Indo-Pacific Ocean should be confirmed. Figure 1b is not enough to support this point.

Following the Reviewer's comment, we have undergone several steps to clarify the extent of the region and quantify its contribution to the global temperature change.

In Section 2.1 (former 3.1), the tropical Indo-Pacific, (S15-N25, as in Section 2.3), is introduced and the contribution of SST warming in this region quantified. Furthermore, Figure 1b) has been adapted to reflect this focus (see revised Manuscript).

We thus added in line 169ff:

"The GMSAT jump, however, results overwhelmingly from the Indo-Pacific Ocean, contributing by about 66% to the GMSAT change, which averages to 0.36°C (29% and 38% from tropical and extratropical Indo-Pacific)."

Even though the tropical Indo-Pacific oceanic domain is limited in surface, its influence on the entire planet is dominant because it is the main source of diabatic heating on average and its variation of the latter explains a large fraction of the global temperature variation (Graham and Barnett, 1987; Brown et al., 2015; Xie et al., 2025).

This justifies the focus on this region throughout our study. To clarify this, we have added following sentence in Section 2.1 (L171ff):

"Tropics and the tropical Indo-Pacific in particular, are the major sources of diabatic heating of the atmosphere (Graham and Bennett, 1987; Zhang et al., 2023) and their variability is thus of great influence for GSAT variations through planetary-scale tropical-extratropical teleconnection (Brown et al., 2015; Xie et al., 2025). We thus focus on the Indo-Pacific region in the following sections."

Furthermore, we have added a paragraph in Section 2.3 to underscore the mechanism and included an analysis of the link between tropospheric temperature and GSAT, for which Figure 4 has been supplemented by panels d) and e) (see reviewed Manuscript).

"The record jump in tropospheric temperature contributes to the jump in GSAT as the released energy within the tropics is distributed by polewards travelling Rossby waves (Alexander et al, 2022). This is confirmed by the linear relation between detrended 500 and detrended GSAT ($R^2 = 0.61$) as shown in Fig.4d. The 2022 to 2023 ASO changes in the two quantities deviates mildly from the regression line towards a larger jump in GSAT than to be expected from the increase in 500 alone points towards. The regression map of the residuals

onto the ASO SST anomalies (Fig.4e) underlines again the importance of the Niña-like SST pattern as well as the Northern Atlantic. “

Nonetheless, we agree that one can not conclusively argue that the described mechanisms are the sole drivers of the observed temperature jump and that in particular land regions show a remarkable temperature change, too. To quantify these changes, we have added an analysis analogous to the one in section 2.1 for global surface air temperature over land (GLSAT). The respective figures has been added to the supplementary materials (Fig.S7a,b, see reviewed Manuscript) and the land contributions as well as other possible factors have been added to the discussion (L607ff):

“Finally, it is very likely that other factors contributed to make the ASO temperature change in 2023, and subsequently for the entire year, so extreme as suggested from Fig.4d. It is indeed important to note that changes in global surface air temperature over land (GLSAT), were record high too in 2023 (Fig. S7a) and contributed about as much as with GMSAT, to the total jump in ASO GSAT global temperature. The land anomalies stem largely from the Tropical and Northern Midlatitudes regions (27% and 42%, respectively, see Fig.S8). Further studies are necessary to investigate and quantify the role of the tropical anomalous heat source through diabatic heating presented here, with respect to other extratropical factors that are responsible for the extreme jump in temperature observed over land (Fig.S7) and ocean (Fig.1a) in the extratropics. For example, there are indications that cloud changes over the northern midlatitudes (Goessling et al., 2024) might have had a contribution, even though they have been following a persistent decade-long trend. In the same vein, the role of the Pacific Decadal Variability in explaining the strong warm anomalies along a broad midlatitude band ranging from 25N to 50N while El Nino was developing in 2023 (Fig.1b) is worth investigating.”

Minor comments:

L25-29: These sentences are not easy to understand. I hope their wording can be improved to make it easier for the readers to grasp the key points, especially the casualties within them.

We have changed the sentence structure to underline the logical chain of the process (L28ff):

“This resulted in (1) a steep year-to-year increase of Sea Surface Temperature (SST), particularly in mean atmospheric subsidence regions , leading to extreme reduction of low cloud cover and giving rise to a record-breaking change in the radiative budget over the central and eastern Indo-Pacific ; (2) anomalous sustained precipitation over climatological

high SSTs in the Western Pacific, fueling unusual diabatic heating and an exceptionally early increase in tropical tropospheric temperature in boreal fall, ultimately influencing the jump GSAT with an additional contribution from the North Atlantic under an extreme state.“

L37: has -> have

Done

L73: (Cattiaux et al.,2024)-> Cattiaux et al.(2024)

Done

L78: El Nino-> El Niño

Done

L81-83: This part is much clearer than L25-29 in the abstract. I suggest the abstract may be revised according to the logic of L25-29.

Abstract is revised and hopefully clearer.

L89: whether “at surface” here is equal to “at 2m”?

Yes, corrected

L90:Some papers indicated that there are some issues with the TOA radiation of ERA5 (e.g., Fig. 1 of Allan and Merchant 2025), which is not consistent with the observation. Allan R. P. and C. J Merchant (2025) Reconciling Earth's growing energy imbalance with ocean warming. Environ. Res. Lett. 20 044002 DOI 10.1088/1748-9326/adb448

The discrepancy seems to affect particularly the recent gradual increase, specifically in the Northern extratropical Pacific (Allan et al., 2025, Fig2). The year to year change in the tropical Indo-Pacific seems to be quite consistent (see Figure S3 of our manuscript and L244f).

L135: The contribution for ... -> The contribution to ...

Obsolete, see next comment

L135-136: It may be revised as :The temperature jump, however, results overwhelmingly from the Indo-Pacific Ocean

Corrected

Figure 1: I was confused about Fig.1a and 1b. The ASO land-masked surface air temperature change between 2023 and 2022 is about 0.58 (roughly estimated). Some explains should be added to clarify the calculation of the difference. Whether it means 2023 minus 2022? While Figure 1b shows that the total oceanic temperature is about 0.28, the difference between 2022 and 2023 should be about 0.4.

Thank you for pointing out what was a figure switch-up.

L164-165: The amplitude may be reduced in the composite result. I suggest “canonical” -> composite

The cross hatching indicates the region where the 2022-2023 change was larger than the change in any single one of the events used for the composite.

Figure 2g-2i: I am curious about the calculation for the composite. I wonder whether the distributions of the red lines here are calculated by the samples (1982, 1987, 1991, 1997, 2009 and 2015) or their average. In addition, the accurate values for the x-axis bins are not clear.

The Mean and Standard Deviation used for the error bars are computed on the combined distribution of all samples, to ensure that extreme cases of other individual years used in the composite are still accounted for.

The data is binned into deciles of the climatological w500 distribution. We don't see an advantage in showing the actual values of the distribution as it is more about its physical value, particularly its sign. The slight offset between composite and 2023 values is only for visualisation purposes, which has been added to the Figure description.

L246: indo-pacific -> Indo-Pacific
Done

Figure S3: El Nino-> El Niño. Please add more explanation about the scatters in Figure S3g-h.

The Figure caption has been updated to read:

“g) Scattered in light grey are monthly standardised anomalies of tropical (S20-N20), indo-pacific TOA radiative budget against tropical indo-pacific low cloud cover along with its linear regression line (dotted) ($R^2 = 0.19$) for the time period of 1979-2023. Arrows indicate the change between two consecutive years: the arrow points from the AMJJAS [-1] average to the AMJJAS [0] average for all the strong El Niños (black) in the composite and 2023 (red). h) as in g) but for tropical indo-pacific low cloud cover against tropical indo-pacific lower tropospheric stability (LTS) with their linear regression shown ($R^2 = 0.29$).”

L327-334: Since the PWS is important for this study. More explanations about the physical meaning of PWS should be added.

We have aimed to improve the explanations of the physical mechanisms behind PWS. For this aim, we have changed the paragraph on the tropospheric temperature anchoring to convective regions in section 2.1 (L228ff) to read:

“Convective adjustment makes the tropospheric temperature (Θ_{Tropo}) follow a moist adiabatic profile, that is anchored to the warmest SSTs in deep convective regions, while dynamic

adjustment renders the temperature horizontally uniform on a timescale of a few days to weeks (Sobel et al., 2002; Bony et al. 2020; Emanuel et al., 1994).”

We refer to back to this in the introductory part of the PWS-mechanism in section 2.2 (L383ff). To motivate the use of PWS (and its advantages over unweighted indices) we have added (L403ff):

“To explain the extremes of free tropospheric temperature that occur during an El-Niño, Sobel et al. (2002) introduced a measure of SST weighted by precipitation, further developed by Flannaghan et al. (2014). The improvements in using precipitation weighted SSTs for tropospheric temperature have been confirmed and used in coupled models (Fueglistaler et al., 2015; Tuel, 2019, Chung et al., 2024) and observational studies (Izumo et al., 2019). [...] “

L385: a Niña-like -> a La Niña-like

Done

L407-409: The causal relationship implied in this sentence is unclear: Consequently, the anomalous SST gradient is located too far east in 2023 (Fig.3h) to efficiently displace or weaken the Walker circulation.

We have changed the sentence to make the logical chain cleared (L503ff):

“In other words, the anomalous SST gradient emerging during El Niño events is located too far east in 2023 (Fig.3h) to efficiently reduce the climatological east-west SST gradient, which canonically leads to a displacement and/or weakening of the Walker cell. “

L413: The sentence is not complete: An exceptional surge in tropospheric warming as early as late-summer/early-fall.

It is part of the enumeration as are the first phrases under point 1. and 2 (and thus a noun phrase). I have changed their capitalisation.

L418: the second point: it transitioned from a cold tropical atmosphere in 2022 set by the preceding La Niña (Fig.4b). Whether this point is also atypical compared to other El Niño event?

We have changed the sentence so that the atypical only refers to the jump, not the initial state, as the Fig.4b (and the detrending method used therein), does indeed not support it (L514ff):

“2023 (i) transitioned from a cold tropical atmosphere in 2022 set by the preceding La Niña (Fig.4b) and (ii) is atypical compared to other El Niños as it is characterized by the strongest tropospheric warming for anomalies of similar amplitude in precipitation-weighted SST, both factors leading to record-high jump in temperature of the tropical troposphere (Fig.S5a).”

References:

Stephens et al. (2015, <https://doi.org/10.1002%2F2014RG000449>)
Loeb et al. (2012, <https://doi.org/10.1007/s10712-012-9175-1>)
Graham and Barnett (1987, <https://doi.org/10.1126/science.238.4827.657>)
Brown et al. (2015, <https://doi.org/10.1002/2014JD022576>)
Xie et al. (2025, <https://doi.org/10.1038/s41612-025-01006-y>)
Alexander et al. (2002, [https://doi.org/10.1175/1520-0442\(2002\)015<2205: TABTIO>2.0.CO;2](https://doi.org/10.1175/1520-0442(2002)015<2205: TABTIO>2.0.CO;2))

We again thank the reviewer for his/her comment which helps considerably improve the original manuscript.

Reviewer #3 (Remarks to the Author):

This paper analyzes the factors behind the extreme global mean surface temperature (GMST) in 2023, with a primary focus on the Indo-Pacific warming. Overall the paper is well-written. While the contribution of the tropical Pacific itself has been discussed in more sophisticated ways in previous studies, including those using climate model ensembles and tropical Pacific pacemaker experiment, this study presents a novel mechanism in which the 2023 El Niño together with La Niña-like background state drives the temperature jump through cloud radiation. This point is interesting and, in my view, merits publication. However, the neglect of the strong midlatitude land warming in 2023 and the lack of quantitative assessment of radiative heating reduce the persuasiveness of this paper as an explanation for the 2023 GMST anomaly. Therefore, I would like to request major revision.

We would like to thank the reviewers for their kind remarks and positive evaluation of the manuscript, as well as the constructive criticism. The reviewer raised important concerns regarding the contribution of the tropical Indo-Pacific to the global temperature jump and the underlying mechanisms of energy redistribution. We have added quantifying analyses and linked them with our findings and extended our discussion.. We are confident that this improves the manuscript. Below are our point-by-point responses to the reviewer's concerns.

Comment 1:

The link between ENSO and radiative fluxes is rather modest (Ceppi and Fueglistaler 2021) in contrast to the well-established strong relationship between ENSO and GMST (Kosaka and Xie 2013; Xie et al. 2025). So it is generally considered that the strong correlation between ENSO and GMST arises mainly through energy redistribution within the Earth. Therefore stronger quantitative evidence is needed to demonstrate that Indo-Pacific radiative heating associated with ENSO significantly impacts local SST and, consequently, GMST.

We agree that the main change in GSAT is explained by the distribution of energy from the tropical oceans. That ENSO has a large impact on radiative fluxes is equally well reported (Kato et al., 2009; Timmermann et al., 2018; Stephens et al., 2015). While the main focus of these studies is often on the variation of long wave radiation, the effect of ENSO on modulating the short wave cloud radiative effect (SW CRE) was already discussed in Loeb et al. 2012, identifying a positive SW CRE for El Ninos and vice versa for La Ninas. These changes were linked to changes in cloud top height and cloud fraction based on CERES observations.

The underlying mechanisms, i.e. the ENSO-driven variability of lower tropospheric stability and low clouds were further reported in observational and modelling studies (Park and Leovy 2004, Radley, Fueglistaler and Donner, 2014).

The radiative effect was both observationally (Flueglistaler et al. 2019) with CERES data as well as identified in Coupled and Atmospheric Climate Models (Ceppi and Flueglistaler, 2021; Wills et al., 2021). Furthermore, the change of low-cloud cover as a response to ENSO related SST changes has been described by Goessling et al., (2024). While Goessling and et al. (2024) argue that the temperature *anomaly* is related to global cloud changes, they also noted that the inter-annual absorbed solar radiation change related to the Nina-to-Nino transition 'largely explain the inconsistencies between the 2023 anomalies and the 2013-2022 trends in the tropics'.

We have strengthened the link between ENSO and radiative fluxes by extending the relevant part of Section 2.2 (L240ff) to read:

“ENSO-driven interannual variability of lower tropospheric stability and low cloud cover, as well as the resulting modulation in shortwave cloud radiative effect, has been reported in observational and modelling studies (Park and Leovy, 2004;, Radley, Fueglistaler and Donner, 2014, Loeb et al., 2012, Wills et al., 2021). Recently, Fueglistaler (2019) used observational records to apply the so-called pattern effect (Stevens et al. 2016; Bloch-Johnson, Rugenstein, and Abbot 2020; Dong et al. 2019; Zhang, Zhao, and Tan 2023) to the build-up of an El Niño event to explain why the warming in the Niño3.4 region is preceded by a several-months-long positive radiative budget anomaly in subsidence regions ; Ceppi and Fueglistaler (2021) subsequently confirmed these results through a study combining models and observations.”

To identify the contribution of the indo-pacific to the ASO temperature change, we compare the periods leading up to ASO 2023 and ASO 2022 as defined in the manuscript. The global mean change TOA of net incoming radiation between these periods is 0.64 W/m² to which the tropical indopacific alone has an area weighted contribution of 0.30 W/m², i.e. 47%. The extremeness of this radiative change is shown in Figure 2j. Furthermore, we have explicitly added in L296ff:

“While many regions throughout the globe have experienced positive radiative anomalies (Goessling et al., 2024), the main contribution of TOA net incoming radiation when averaged globally between consecutive AMJJAS seasons, lies in the tropical Indo-Pacific with an area-weighted contribution of 47%. The resulting upper ocean warming in the tropics sets the scene for precipitation and atmospheric circulation changes and related processes, which are examined subsequently. “

For change in low clouds the area weighted contribution of the tropical Indo-Pacific to the global low cloud decrease is similarly important at 51%.

For the reviewer's interest, but not included in the manuscript, this Figure shows the change of radiation in the month leading up to the ASO season as in Fig.2j for the entire globe without the tropical Indo-Pacific. (Note the different scale, as mean changes in the tropical Indo-Pacific are significantly larger than global averages). The 2022 to 2023 still is at the 74th percentile of the full 1979-2024 distribution and larger than most comparable El-Ninos, but smaller than the 1996-1997 change. While radiation outside of the tropical Indo-Pacific is thus also important for the observed temperature change, the Indo-Pacific plays a major part.

Figure R1. Change in Global Net TOA Radiation with Tropical Indo-Pacific excluded. Change of global, except tropical (S20-N20), indo-pacific, average of TOA radiative Budget between AMJJAS [-1] and AMJJAS [0] for the years 1979-2024 (grey), strong El-Niños (black) and 2023 (red) in W/m-2 from ERA5.

As the tropical Indo-Pacific radiation change has been largely attributed to changes in low clouds, the additional energy is either deposited directly in the lower atmosphere, where it influences local air temperature or in the tropical upper ocean, from where it influences global surface air temperature through the processes described in Section 2.3. On this point, we have added in the discussion in line 544ff:

“While these processes are originally locally confined to the Indo-Pacific, their impacts, mediated through the diabatic heating and the constraints of weak temperature gradients, are global (Fig.4d). They contribute to explain the extraordinary temperature changes observed in the extratropical Pacific (Fig.1b) and over land.”

Lastly, we agree that it is crucial to quantify how much these two processes bring in extra energy exactly. We do not think that this is possible from the limited observational record and hope this reflects in our manuscript final sentence (L621ff):

“Analyses of long preindustrial control simulations beyond CMIP6 historical runs, completed by dedicated model sensitivity experiments are required to isolate and quantify the role of the different internally-driven processes documented here, as well as their interaction with the mean background state and would help to deepen our understanding of the observed 2023 jump in global temperature.”

Comment2: The 2023 GMST jump was pronounced over extratropical land regions (Xie et al. 2025). However, this study focuses exclusively on sea surface temperature, which makes the explanation for the GMST jump incomplete. Tropical Pacific SST pacemaker experiment reproduces tropical surface warming reasonably well but fails to capture extratropical warming (Xie et al. 2025). This inconsistency suggests that the GMST jump cannot be fully explained by the tropical effect.

We think that this is an important remark and have addressed it by a) quantifying the direct contribution of the tropical Indo-Pacific to the GSAT jump, b) providing quantitative evidence how the tropical Indo-Pacific impacts extratropical regions and c) quantifying and discussing the contribution from the land areas.

In Section 2.1, the tropical Indo-Pacific, (S15-N25, as in Section 2.3), is introduced and the contribution of SST warming in this region quantified. Furthermore, Figure 1b) has been adapted to reflect this focus (see revised Manuscript). We thus added in line 169ff:

“The GMSAT jump, however, results overwhelmingly from the Indo-Pacific Ocean, contributing by about 66% to the GMSAT change, which averages to 0.36°C (29% and 38% from tropical and extratropical Indo-Pacific).”

Even though the tropical Indo-Pacific oceanic domain is limited in surface, its influence on the entire planet is dominant because it is the main source of diabatic heating on average and its variation of the latter explains a large fraction of the global temperature variation (Graham and Barnett, 1987; Brown et al., 2015; Xie et al., 2025).

This justifies the focus on this region throughout our study. To clarify this, we have added following sentence in Section 2.1 (L171ff):

“Tropics and the tropical Indo-Pacific in particular, are the major sources of diabatic heating of the atmosphere (Graham and Bennett, 1987; Zhang et al., 2023) and their variability is thus of great influence for GSAT variations through planetary-scale tropical-extratropical teleconnection (Brown et al., 2015; Xie et al., 2025). We thus focus on the Indo-Pacific region in the following sections.”

Furthermore, we have added a paragraph in Section 2.3 (L460ff) to underscore the mechanism and included an analysis of the link between tropospheric temperature and GSAT, for which Figure 4 has been supplemented by panels d) and e) (see reviewed Manuscript)

“The record jump in tropospheric temperature contributes to the jump in GSAT as the released energy within the tropics is distributed by polewards travelling Rossby waves (Alexander et al, 2022). This is confirmed by the linear relation between detrended 500 and detrended GSAT ($R^2 = 0.61$) as shown in Fig.4d. The 2022 to 2023 ASO changes in the two quantities deviates mildly from the regression line towards a larger jump in GSAT than to be expected from the increase in 500 alone points towards. The regression map of the residuals onto the ASO SST anomalies (Fig.4e) underlines again the importance of the Niña-like SST pattern as well as the Northern Atlantic.”

Nonetheless, we agree that one can not conclusively argue that the described mechanisms are the sole drivers of the observed temperature jump and that in particular land regions show a remarkable temperature change, too. To quantify these changes, we have added an analysis analogous to the one in section 2.1 for global surface air temperature over land (GLSAT). The respective figures has been added to the supplementary materials (Fig.S7a,b, see reviewed Manuscript) and the land contributions as well as other possible factors have been added to the discussion (L607ff):

“Finally, it is very likely that other factors contributed to make the ASO temperature change in 2023, and subsequently for the entire year, so extreme as suggested from Fig.4d. It is indeed important to note that changes in global surface air temperature over land (GLSAT), were record high too in 2023 (Fig. S7a) and contributed about as much as with GMSAT, to the total jump in ASO GSAT global temperature. The land anomalies stem largely from the Tropical and Northern Midlatitudes regions (27% and 42%, respectively, see Fig.S8). Further studies are necessary to investigate and quantify the role of the tropical anomalous heat source through diabatic heating presented here, with respect to other extratropical factors that are responsible for the extreme jump in temperature observed over land (Fig.S7) and ocean (Fig.1a) in the extratropics. For example, there are indications that cloud changes over the northern midlatitudes (Goessling et al., 2024) might have had a contribution, even though they have been following a persistent decade-long trend. In the same vein, the role of the Pacific Decadal Variability in explaining the strong warm anomalies along a broad midlatitude band ranging from 25N to 50N while El Nino was developing in 2023 (Fig.1b) is worth investigating.”

Lastly, we would like to directly respond to the reviewers remark that these extratropical connections are not captured in the pacemaker experiments of Xie et al.

Xie et al. 2025 show that the extratropical temperatures cannot be explained through the SSTs in the Nino3.4 region. Our method yields similar results: The Detrended Nino3.4 Index correlates with the detrended Extratropical Indo-Pacific significantly less well ($R^2=0.31$) and the change of these two quantities from 2022 to 2023 does not follow the regression line. This is in line with our argument, that it is the SSTs in regions of deep convection that are the most important. When relating the detrended Extratropical Temperatures to the detrended PWS anomaly, the correlation increases significantly ($R^2=0.70$) and the 2022-2023 jump is more in line with the regression.

Comment3:

The connection between the tropospheric warming discussed in Section 3.3 and GMST is not clear. In relation to *Section 3.2, the tropospheric warming could increase low clouds*. It would be helpful if the role of Section 3.3 could be clarified.

We hope that the answer and manuscript changes in response to comment 1 have made the role of section 2.3 clear and that the following paragraph helps to make the connection between the section clear (L544ff):

“While these processes are originally locally confined to the Indo-Pacific, their impacts, mediated through the diabatic heating and the constraints of weak temperature gradients, are global (Fig.4d). They contribute to explain the extraordinary temperature changes observed in the extratropical Pacific (Fig.1b) and over land.”

Concerning the impact of tropospheric warming on the low clouds: Indeed, the early increase in lower tropospheric warming restores the lower tropospheric stability faster than during other El Ninos. The importance here lies in the timing of the two different processes. While the initial decrease in LTS and LCC and the resulting increase in radiation takes place in the month leading up to the ASO season (for which we have chosen AMJJAS, following Ceppi and Fueglistaler, 2021) the increase of tropospheric warming starts around JJA.

From an initially very stable troposphere, the LTS decreases strongly in the month leading up to the ASO season, in line with the increase of local SSTs (see Fig. 1c). Shortly before the ASO season, in line with increasing tropospheric temperatures (see Fig.4a), the *tropically averaged* stability is again restored. The low cloud cover is related to the LTS mostly in regions of subsidence.

We have included a comment on this in the discussion section (L541ff):

“ASO is a pivotal season in the Indo-Pacific because it corresponds to the moment of the year that is influenced by the radiative anomalies developed a few months earlier over the basin and marks the beginning of the increase in diabatic heating and thus tropospheric temperature. “

Comment4:

From the title and abstract, the paper seems to study the 2023 temperature. But the paper mostly focuses on the process for the Indo-Pacific changes in ASO. It is unclear if this picture contributes to the annual mean temperature.

As shown by Cattiaux et al. (2024), the ASO season is crucial in rendering 2023 extreme. For the reviewers interest, but not included in the manuscript, the Figures provided below show the change of global surface air temperature for between subsequent years for annual means (a) and for annual means with the ASO season excluded. Performing the same analysis for global marine surface air temperatures makes the difference even more apparent, highlighting the potential role of land regions as a secondary effect, as discussed in the answer to comment 2.

a)

Figure R2. Global Surface Air Temperature Change for the whole year and ASO season excluded. a) Change of GSAT between consecutive years for the 1979-2024 (grey), with strong El-Niño years highlighted in black (dashed for years of volcanic eruptions) and 2023 in red, all in °C (ERA5). b) As in a) with the ASO season excluded.

We have made the studies focus on the ASO season and its justification more explicit in line 111ff:

“Our study uses observations to connect the specific characteristics of the 2023 El-Niño with the observed extreme jump in the early fall (ASO) GSAT and subsequent annual record, based on a process-based approach using observations only. We will focus on that specific early fall season which is considered as atypical for an ENSO year following Cattiaux et al. (2024).”

Specific comments:

L157: "Radiative forcing" sounds like external forcing. Perhaps it should be rephrased. Rephrased to radiative budget.

L184: typo
Corrected

L205: Despite the extreme coastal Niño, low cloud change along the Peruvian Coast looks small.

We have no clear explanation for this feature, which appears to be consistent in radiation, low cloud and high cloud cover. We expect this to be a small scale effect with a negligible impact on the net radiative effect.

L231: LTS has already been defined

Corrected

L233: Suggest splitting this long sentence

The sentence was removed as it no longer fit well into this section.

L308: remove ","

Done

Fig. S6: typo in the title

Done

Reference:

Ceppi and Fueglistaler (2021, <https://doi.org/10.1029/2021GL095261>)

Kosaka and Xie (2013, <https://doi.org/10.1038/nature12534>)

Xie et al. (2025, <https://doi.org/10.1038/s41612-025-01006-y>)

References:

Cattiaux et al. (2024, <https://doi.org/10.1029%2F2024GL110531>)

Kato et al (2009, <https://doi.org/10.1175/2009JCLI2795.1>)

Timmermann et al (2018, <https://doi.org/10.1007/s10712-012-9175-1>)

Stephens et al. (2015, <https://doi.org/10.1002%2F2014RG000449>)

Loeb et al. (2012, <https://doi.org/10.1007/s10712-012-9175-1>)

Graham and Barnet (1987, <https://doi.org/10.1126/science.238.4827.657>)

Brown et al. (2015, <https://doi.org/10.1002/2014JD022576>)

Xie et al. (2025, <https://doi.org/10.1038/s41612-025-01006-y>)

Alexander et al. (2002,

[https://doi.org/10.1175/1520-0442\(2002\)015<2205:TABTIO>2.0.CO;2](https://doi.org/10.1175/1520-0442(2002)015<2205:TABTIO>2.0.CO;2))

Wills et al. (2021, <https://doi.org/10.1029/2022GL100011>)

We again thank the reviewer for his/her comment which helps considerably improve the original manuscript.

Reply to Reviewers

REVIEWERS' COMMENTS:

Reviewer #1 (Remarks to the Author):

I thank the authors for addressing my relatively modest comments. On my behalf, I can now recommend publication. Please note that there is a brand new paper on September 2023 global jump which you may want cite in your paper:

<https://www.nature.com/articles/s43247-026-03178-8>

We thank Rev#1 for their recommendation. The reference was included in the introduction (L63).

Reviewer #2 (Remarks to the Author):

It is a pleasure to review this manuscript again. Following the authors' revisions, I now have a better understanding of the entire work, and I only have a few minor comments and suggestions.

We thank Rev#2 for their replies and their comments. Below are our point-by-point responses to the Rev#2's new comments and suggestions.

L88: El-Niño -> El Niño

Thank you, corrected.

L119-120: We thus focus on the Indo-Pacific region in the following sections. -> We thus focus on the tropical Indo-Pacific region in the following sections.

Thank you, corrected.

L228-231: Based on my understanding, Figure 2 indicates that changes in low clouds over certain regions still follow the constraining relationship of LTS, but this relationship may not hold at the regional average level. Is my understanding correct?

Yes, that is correct

L312: ofanomalous ->of anomalous

Thank you, corrected.

L341-342:If the relationship between delta PWS and delta theta is directly calculated, would it better illustrate the conclusions of the paper without the need for detrending? It's just a suggestion.

We agree with the reviewer's suggestions and it was indeed a consideration that we had at one point during our writing and story-telling process. The Figures below, attached solely for the Reviewer's interest, show the year to year change in PWS and potential temperature and the residual correlation, (same as in Figure 3b) and c), but for the year-to-year changes.

While the year 2023 stands out as more extreme in this view point, the perspective of the preconditioning, i.e. that it is starting from an anomalously low state in PWS and Theta_500, is lost. This is why we decided to use the detrended values instead.

Figure Rev#2.1 Changes in Tropospheric Warming. a) As Figure 4b), but for year-to-year changes in tropical (15S-25N), tropospheric potential temperature for the ASO season, against year-to-year changes in precipitation-weighted sea surface temperature (PWS). **b)** As Figure 4c) but for the residuals of a).

L392: Niña-like-> La Niña-like

Thank you, corrected.

L476-477: The spatial pattern of the El Niño events (double centers) seems to be a factor to the global surface temperature (Geng et al. 2024; Jiang et al. 2025).

See next comment.

L529-531: Recent studies have quantified the contributions of different basins to GSAT and GLSAT(Jiang et al. 2025).

Thank you for this reference. Regarding the contribution of the different basins, the Simple Green's Function Model proposed in this paper complements well our argument and underlines the importance of the Western Pacific for extratropical regions. We have included it in Line 540:

'While statistical models using the Green's Function Method highlight the importance of the central and western Pacific (Jiang et al., 2025), further studies are necessary to investigate and quantify the role of the tropical anomalous heat source through diabatic heating presented here, with respect to other factors that are responsible for the extreme jump in temperature observed over land (Fig.S1) and ocean (Fig.1a) in the extratropics. [...]

For Jiang et al. focusing on the time period from July 2023 to June 2024, the double peak structure is crucial. With the central peak maturing around the end of the year, it does not

appear to be a driver for the key mechanisms of our work, which make the year 2023 stand out.

Reference:

Geng, X., Kug, JS., Shin, NY. et al. On the spatial double peak of the 2023–2024 El Niño event. *Commun Earth Environ* 5, 691 (2024). <https://doi.org/10.1038/s43247-024-01870-1>

Jiang, N., Zhu, C., McPhaden, M.J. et al. Atypical warming pattern of strong 2023-24 El Niño boosts global temperatures to new 1.5 °C record. *Commun Earth Environ* 6, 1012 (2025). <https://doi.org/10.1038/s43247-025-02971-1>

Reviewer #3 (Remarks to the Author):

This is the second review of the manuscript. I thank the authors for carefully addressing the previous comments. I have several comments, which I believe can be addressed by modifying the text.

We thank the Rev#3 for his/her reply and thoughtful comments and suggestions, which, yet again, have helped improve the manuscript. Below we provide a point-by-point response to the new concerns.

Section 2.3: This section starts with a detailed discussion of precipitation and tropospheric heating, whose roles in GSAT have not been clearly mentioned earlier in the manuscript. This is a point where readers may easily get lost. To guide readers, it would be helpful to clarify at the beginning of the subsection how these processes are linked to the previous subsection and how they contribute to the GSAT jump.

We agree that the proposed restructuring helps clarify the argument. We have thus moved the following paragraph to the beginning of section 2.3:

‘As outlined in the introductory part of section 2.2, it is the SST in regions of deep convection, that control a large fraction of variance in tropospheric temperature through injection of anomalous diabatic heating into the tropical atmospheric column; from there, the released latent heat influences GSAT through teleconnections. To understand the temporal evolution of tropospheric temperature, we assess the spatio-temporal evolution of precipitation anomalies [...]

In Line 311, where it was removed, after the description of the precipitation and SST specificities of 2023, we added:

‘As a result of these atypical precipitation-to-SST configurations, the tropical atmospheric warming in 2023 started earlier [...]

Section 3: The latter part of this section (L469-) is excessively long and involved, despite being speculative. I suggest shortening this part.

We think that this section is important to put our work into connection with existing lines of research, to point out potential ways to move forward in further studies and to contribute to the actual hot debate about why models might struggle to simulate the event. Furthermore, here we address several remarks raised by the other reviewers in the first round of reviews about additional potential processes contributing to the temperature jump.

Nonetheless, we have shortened the section by moving the result of the land analysis to the section 2.1 and by removing the following paragraphs, accordingly:

L499ff:

'It is worth noting that the Pacific Decadal Oscillation, which is very well correlated with IPV, remains significantly negative across the full La Niña–El Niño 2022-2024 cycle as part of a long-lasting multidecadal negative phase initiated in early 2000s and still ongoing (NOAA, <https://www.ncei.noaa.gov/access/monitoring/pdo/>).'

L534ff:

'It is indeed important to note that changes in global surface air temperature over land (GLSAT), were record high too in 2023 (Fig. S7a) and contributed about as much as with GMSAT, to the total jump in ASO GSAT global temperature. The land anomalies stem largely from the Tropical and Northern Midlatitudes regions (27% and 42%, respectively, see Fig.S7).'

L544ff:

'In the same vein, the role of the Pacific Decadal Variability in explaining the strong warm anomalies along a broad midlatitude band ranging from 25°N to 50°N while El Niño was developing in 2023 (Fig.1b) is worth investigating.'

Analysis of GLSAT should be in the Results section.

We have moved the result of the GLSAT analysis into the Result section 2.1, Line 116:

'The GMSAT jump, however, results overwhelmingly from the Indo-Pacific Ocean, contributing by about 66% to the GMSAT change, which averages to 0.36°C (29% and 38% from tropical and extratropical Indo-Pacific, respectively). Furthermore, the changes in global surface air temperature over land (GLSAT), were record high too in 2023 (Fig. S7a) and contributed about as much as with GMSAT, to the total jump in ASO GSAT global temperature. The land anomalies stem largely from the Tropical and Northern Midlatitudes regions (27% and 42%, respectively, see Fig.S7). Tropical oceans, and the tropical Indo-Pacific in particular, are the major sources of diabatic heating of the atmosphere (Graham and Benett, 1987; Zhang et al., 2023) and their variability is thus of great influence for GSAT variations through planetary-scale tropical-extratropical teleconnection (Brown et al., 2015; Xie et al., 2025). We thus focus on the Indo-Pacific region in the following sections.'

L386: It is not obvious that tropical heating drives extratropical warming. In fact, the ENSO influence on GSAT is dominated by surface warming in the tropics (Wang et al., 2025; Xie et al., 2025). The high correlation between GSAT and tropical tropospheric temperature is likely tied to surface warming in the tropics, rather than extratropical responses.

Indeed, ENSO does not significantly and directly impact extratropical Pacific (Xie et al.; Panel a), but the PWS (linked to tropospheric heating) does indirectly, as shown in the following Figure Rev3 (not included in the paper but for Rev3 only) for the extratropical Indopacific domain as defined in Figure 1b (Panel b). This is in line with the corps of literature on Green Function Analysis, reporting a strong and robust warming of the whole free troposphere that manifests in global surface air temperature warming, including in extratropical regions, as a response to SST warming in tropical ascent regions (e.g. Zhou et al., 2017, Dong et al., 2019, Zhang et al., 2023). Similarly, PWS does explain more than 40% of variance of Midlatitudinal warming over land as defined in Fig.S7b (see Fig Rev3.1c-g).

Figure Rev3.1 SST-weighted Precipitation Influences on Extratropical Temperature.

a) As Figure 4 b) but for detrended extratropical indopacific Surface air temperature anomaly against the detrended Nino3.4 anomaly ($R^2 = 0.32$). **b)** As Figure 4b) but for detrended extratropical indopacific surface air temperature anomaly against the detrended precipitation-weighted sea surface temperature (PWS) ($R^2 = 0.70$). **c)-g)** As for Figure 4b) and c) but for Latitudinal bands of GLSAT, as defined in Figure S7b. ($R^2 = \{0.41, 0.42, 0.20, 0.42, 0.45\}$).

In all cases, for the extratropical Indo-Pacific region as well as for all other land areas, the temperature jump of 2023 is larger than what would be expected from the linear regression,

pointing towards non-linearities or further processes. The residual correlation maps are all reminiscent of La-Nina like conditions in the Pacific and also suggest a strong impact of the tropical North Atlantic. The potential non-linearities due to this pattern as well as further processes that could be involved, are discussed in Section 3 L514ff. We also stress the necessity to conduct experiments to quantify the role of these processes, as pointed out by Rev3 in their previous comments.

L118: As noted above, tropical influence on extratropical temperature is not the major factor for GSAT variability.

See answer provided above.

Fig. 4: There are two panels labeled (d).

Thank you, corrected.

L356: Typo.

Thank you, corrected.

L388: Alexander et al., 2002.

Thank you, corrected.

L442: Item 4 does not exist.

Thank you, delete.

L465: I don't see any evidence for this.

See answer provided above.

L580, and some other references: Incorrect indent.

Thank you, corrected.

References:

Wang et al. 2017: Global Influence of Tropical Pacific Variability with Implications for Global Warming Slowdown. <https://doi.org/10.1175/JCLI-D-15-0496.1>

Xie et al. 2025: What made 2023 and 2024 the hottest years in a row? <https://doi.org/10.1038/s41612-025-01006-y>

References:

Zhou, C., M. D. Zelinka, and S. A. Klein (2017), Analyzing the dependence of global cloud feedback on the spatial pattern of sea surface temperature change with a Green's function approach, J. Adv. Model. Earth Syst., 9, 2174–2189, doi:10.1002/2017MS001096.

Dong, Y., C. Proistosescu, K. C. Armour, and D. S. Battisti, 2019: Attributing Historical and Future Evolution of Radiative Feedbacks to Regional Warming Patterns using a Green's Function Approach: The Preeminence of the Western Pacific. *J. Climate*, 32, 5471–5491, <https://doi.org/10.1175/JCLI-D-18-0843.1>

Zhang, B., M. Zhao, and Z. Tan, 2023: Using a Green's Function Approach to Diagnose the Pattern Effect in GFDL AM4 and CM4. *J. Climate*, 36, 1105–1124, <https://doi.org/10.1175/JCLI-D-22-0024.1>.